



# Causes of interannual variability of tropospheric ozone
# over the Southern Ocean
Junhua Liu[1,2], Jose M. Rodriguez[2], Stephen D. Steenrod[1,2], Anne R. Douglass[2], Jennifer
A. Logan[3], Mark Olsen[2,4], Krzysztof Wargan[2,5], Jerald Ziemke[2,4]
[1] Universities Space Research Association (USRA), GESTAR, Columbia, MD, USA
[2] NASA Goddard Space Flight Center, Greenbelt, Maryland, USA
[3] School of Engineering and Applied Sciences, Harvard University, Cambridge, MA,
USA
[4] Morgan State University, Baltimore, Maryland, USA
[5] Science Systems and Applications, Inc., Lanham, MD, USA
*Correspondence to*: Junhua Liu (junhua.liu@nasa.gov)



**Abstract.**
We examine the relative contribution of processes controlling the interannual variability
(IAV) of tropospheric ozone over four sub-regions of the southern hemispheric
tropospheric ozone maximum (SHTOM) over a twenty-year period. Our study is based
on hindcast simulations from the National Aeronautics and Space Administration Global
Modeling Initiative – Chemistry transport model (NASA GMI-CTM) of tropospheric and
stratospheric chemistry, driven by assimilated Modern Era Retrospective-Analysis for
Research and Applications (MERRA) meteorological fields. Our analysis shows that over
SHTOM region, the IAV of the stratospheric contribution is the most important factor
driving the IAV of upper tropospheric ozone (270 hPa), where ozone has a strong
radiative effect. Over the south Atlantic region, the contribution from surface emissions
to the IAV of ozone exceeds that from stratospheric input at and below 430 hPa. Over the
south Indian Ocean, the IAV of stratospheric ozone makes the largest contribution to the
IAV of ozone with little or no influence from surface emissions at 270 hPa and 430 hPa
in austral winter. Over the tropical south Atlantic region, the contribution from IAV of
stratospheric input dominates in austral winter at 270 hPa and drops to less than half but
is still significant at 430 hPa. Emission contributions are not significant at these two
levels, even during September. The IAV of lightning over this region also contributes to
the IAV of ozone in September and December. Over the tropical southeastern Pacific, the
contribution of the IAV of stratospheric input is significant at 270 hPa and 430 hPa in
austral winter, and emissions have little influence.
**1 Introduction**
Tropospheric ozone plays a critical role in controlling the oxidative capacity of the
troposphere through its photolysis in the presence of water vapor, generating hydroxyl
radical (OH), the main atmospheric oxidant (e.g., Logan et al., 1981). It contributes to
smog and is harmful to human and ecosystem health near the surface. It acts as a
greenhouse gas in the upper troposphere (Lacis et al., 1990) and affects the radiative
forcing of the climate system. Tropospheric ozone is produced by photochemical
oxidation of CO and volatile organic compounds (VOCs) in the presence of nitrogen



oxides ($NO_X$) (e.g., Logan et al., 1981). Downward transport of ozone from the
stratosphere is also an important source of tropospheric ozone (e.g., Danielsen, 1968;
Stohl et al., 2003). Deep convection and long-range transport of ozone and its precursors
also modulate the tropospheric $O_3$ distributions (e.g., Chandra et al., 2009; Oman et al.,

46    2011).

Our study is motivated by the existence of tropospheric ozone maximum over tropical
and subtropical southern hemisphere as seen both in model simulations and GMAO
assimilated ozone product derived from OMI/MLS satellite measurements (Figure 1).
Although in the southern hemisphere tropospheric air is relatively "clean" and less
polluted compared with the Northern Hemisphere, this tropospheric ozone column
maximum reaches as high as 35DU and is comparable to the typical northern mid-latitude
values of 30DU. The elevated tropospheric ozone column is centered over the south
Atlantic from the equator to 30°S, and is part of the well-known tropical wave-one
pattern first noted in observations made by the Nimbus 7 Total Ozone Mapping
Spectrometer (TOMS) (e.g., Fishman et al., 1990; Ziemke et al., 1996). This ozone
maximum extends westward to South America and the tropical southeastern Pacific,
southeastward to southern Africa, south Indian Ocean along the latitude band of 30°S-
45°S, and is a dominant global feature (Thompson et al., 2003; Sauvage et al., 2007).
This elevated ozone region exists year-around, with a seasonal maximum in August -
October, and a seasonal minimum in April - May.
This study provides an examination of the relative contributions of the factors that control
the interannual variations of the southern hemisphere tropospheric ozone maximum over
a twenty-year period. Prior studies have examined the processes that produce the
southern hemisphere tropospheric ozone maximum (SHTOM), but consider only short
periods or are limited in spatial scale. These studies concluded that horizontal and vertical
transport of ozone precursors from regions of biomass burning (e.g., Jacob et al., 1996;
Pickering et al., 1996; Thompson et al., 1996; Jenkins and Ryu, 2004a; Sauvage et al.,
2006; Jourdain et al., 2007; Thouret et al., 2009), lightning NOx (Martin et al., 2002;
Jenkins and Ryu, 2004b; Kim et al., 2013; Tocquer et al., 2015) and stratospheric
intrusions (Weller et al., 1996) all contribute to this tropospheric ozone column
maximum. However, changes of the relative contributions of these factors to tropospheric



ozone on inter-annual time scale over this region have not been examined in detail.
Studies considering tropospheric ozone interannual variability have not focused on the
SHTOM region. Zeng et al. (2005) used a combined climate/chemistry model to evaluate
the ENSO effects on the interannual variability of tropospheric ozone. Their study
concludes that STE variation induced by ENSO is one important factor driving the IAV
of the global mean of tropospheric ozone. Vouldgarakis et al. (2010) examined the
drivers of interannual variability of the global tropospheric ozone using the p-TOMCAT
tropospheric chemistry transport model (CTM). Their study shows that changing
transport including the STE is important in determining the IAV of tropospheric ozone.
The influence of emissions is confined to areas of intense burning on the interannual
timescale. Murray et al. (2013) examined the effects of lightning on the IAV in the
tropical tropospheric ozone column based on the GEOS-Chem CTM with IAV in tropical
lightning constrained by satellite observations from Lightning Imaging Sensors (LIS).
Their study finds that lightning plays an important role in driving the IAV of tropical
tropospheric ozone column, especially over East Africa, central Brazil, and in continental
outflow in the eastern Pacific and the Atlantic, but their model does not reproduce the
IAV in TCO except in East Africa and central Brazil. Liu et al. (2016) analyzed
simulations from a global chemistry and transport model to show that the IAV in the
stratospheric contribution significantly affects the IAV of upper tropospheric ozone at the
SHADOZ station over Reunion (21°S). In this study, we focus on the SHTOM region
and quantify the relative contributions of several factors to the tropospheric ozone
interannual variability during the past twenty years. We examine the horizontal and
vertical variations of these contributions by separating the SHTOM into four subregions
and comparing their IAVs at two selected levels (270 hPa and 430 hPa). This analysis
distinguishes between anthropogenic and natural sources on the IAV of the tropospheric
ozone and their contributions to the radiative forcing changes.
In this study, we use a global chemistry transport model to identify the processes
impacting observed interannual variability of the tropospheric ozone column maximum in
southern hemisphere. We examine the model sensitivity of tropospheric ozone to
different ozone sources through the use of multiple linear regression. We include
stratospheric input and emissions as two major predictor variables in our regression. In



our GMI-CTM, the global total of $NO_X$ from lightning is fixed at 5.0 TgN/yr. The
regional $NO_X$ emission from lightning is coupled to the deep convective transport and
mixing in the model varies from year to year. We include the lightning $NO_X$ as the third
factor in our regression model over the tropical south Atlantic region, where ozone is
sensitive to the IAV of lightning $NO_X$ as found in Murray et al (2013). In our multiple
linear regression, a regression coefficient that is significantly different from zero at the
95% confidence level implies that the corresponding process contributes significantly to
the variation of simulated ozone. We thereby quantify their relative contributions to the
interannual variability of tropospheric ozone. Our study focuses on austral winter season
when the subtropical jet related stratosphere - troposphere exchange reaches the seasonal
maximum (Karoly et al., 1998; Bals-Elsholz et al., 2001; Nakamura and Shimpo, 2004).
Southern hemisphere biomass burning (e.g., Liu et al., 2010; 2013) also reaches the
maximum during this season.
Section 2 briefly describes the model and simulations, including the standard chemistry
simulation, the stratospheric $O_3$ tracer simulation, and the tagged CO simulation. It also
describes GEOS-5 ozone assimilation, as the assimilated fields are used to evaluate
model performance over the southern hemisphere extra-tropics and tropics as discussed
in the first part of section 3. The second part of section 3 presents a diagnostic study of
controlling factors, including stratosphere input, surface emissions and lightning, on the
tropospheric ozone IAV relying on a series of hindcast simulations from 1992 to 2011.
Section 4 is a summary and conclusion.
**2 Model and Data**
**2.1 Model**
We used the Global Modeling Initiative chemical transport model (GMI-CTM) (Duncan
et al., 2007; Strahan et al., 2007), driven by MERRA reanalysis meteorology (Rienecker
et al., 2011, http:// gmao.gsfc.nasa.gov/research/merra/). The native resolution of the
MERRA field is 0.67° × 0.5° with 72 vertical levels; we regrid it to 2°x2.5° for input to
the GMI-CTM simulations in this study.



The chemical mechanism used in GMI-CTM represents stratospheric and tropospheric
chemistry with offline aerosols input from GOCART model simulations (Chin et al.,
2002). The GMI-CTM hindcast simulation has been used and compared to observations
in many recent studies. Strahan et al. (2013) showed excellent agreement between
simulated and MLS ozone profiles in the Arctic lower stratosphere and. Liu et al. (2016)
shows the GMI-CTM hindcast and ozonesonde agree very well on the annual cycles and
IAV over Reunion from lower troposphere to the upper troposphere. Strode et al. (2015)
shows that the GMI-CTM hindcast reproduces the seasonal cycle and IAV of observed
surface ozone over United States from Environmental Protection Agency (EPA)'s Clean
Air Status and Trends Network (CASTNET).
The sources of emissions in the GMI-CTM standard simulation are summarized in the
recent study of Strode et al. (2015). Besides the standard simulation, we also carry out a
control run with constant emissions fixed at the year 2000 levels to quantify effects of
emission IAV on ozone IAV.
The lightning parameterization in the model follows the scheme described by Allen et al
(2010) and the emissions of lightning $NO_x$ are calculated online. The global total of $NO_x$
from lightning is fixed every year but varies spatially.
Methane mixing ratios are specified in the two lowest model levels, using time dependent
zonal means from National Oceanic and Atmospheric Administration / Global
Monitoring Division (NOAA/GMD). Other long-lived source gases important in the
stratosphere, such as $N_2O$, CFCs, halocarbons are prescribed at the two lowest model
levels following the A2 scenario by (WMO, 2014). Stratospheric aerosol
distributions/trends are from International Global Atmospheric Chemistry/Stratospheric
Processes And their Role in Climate (IGAC/SPARC) and have IAV (Eyring et al., 2013).
The model includes a stratospheric $O_3$ tracer (StratO$_3$). The StratO$_3$ is defined relative to a
dynamically varying tropopause tracer (e90) (Prather et al., 2011). This artificial tracer is
set to a uniform mixing ratio (100 ppb) at the surface with 90 days e-folding lifetime. In
our simulation, the e90 tropopause value is 90 ppb. The StratO$_3$ tracer is set equal to $O_3$ in
the stratosphere and is removed in the troposphere with the same loss frequency
(chemistry and deposition) archived from daily output of the standard chemistry model
simulation with yearly-varied emission in this study. Using the StratO$_3$ tracer allows



quantification of $O_3$ of stratospheric origin in the troposphere at a given location and
time. This approach has also been adopted in the high resolution GFDL AM3 model (Lin
et al., 2012).
In this study, we also conducted a tagged CO simulation to examine the emission sources
during the same period as the full chemistry simulation. The tagged CO simulation has
horizontal resolution of 1°x1.25°. The primary chemical loss of CO is through reactions
with OH radicals, which are archived from the respective standard chemistry simulation
with yearly-varied emissions. The chemical production and loss rates of CO in the
stratosphere were archived from the respective standard chemistry simulations.
**2.2 GMAO GEOS-5 Ozone Assimilation**
We used assimilated tropospheric ozone to evaluate model performance. This assimilated
dataset is produced by ingesting OMI v8.5 total column ozone and MLS v3.3 ozone
profiles into a version of the Goddard Earth Observing System, Version 5 (GEOS-5) data
assimilation system (Rienecker et al., 2011). No ozonesonde data are used in the
assimilation. Wargan et al. (2015) provides details of the GEOS-5.7.2 assimilation
system, which for this application is produced with 2° x 2.5° horizontal resolution and
with 72 vertical layers between the surface and 0.01 hPa. For the troposphere, the
assimilation only applies a dry deposition mechanism at the surface without any chemical
production or loss. This algorithm works since the ozone lifetime is much longer than the
six-hour analysis time on which the background field is corrected by observations.
Ziemke et al. (2014) evaluated the tropospheric ozone profiles derived from three
strategies based on OMI and MLS measurements, including this GEOS-5 assimilation,
trajectory mapping and direct profile retrieval using residual method, with ozonesonde
observations and GMI model simulations. They show that the ozone product (500 hPa to
tropopause) from the GEOS-5 assimilation is the most realistic. Wargan et al. (2015) also
demonstrate that the ozone between 500 hPa and the tropopause from GEOS-5
assimilation is in good agreement with independent observations from ozonesondes. The
assimilation applies the OMI averaging kernels in the troposphere, but the weight of OMI
kernels decreases sharply below 500 hPa (Personal communication with K. Wargan).
Considering that in the lower troposphere there is no direct observational constraint in





193 the analysis, we use ozone mixing ratio at 270 hPa and 430 hPa as well as partial column

194 ozone integrated from 500 hPa to the tropopause from GEOS-5 assimilation as a

195 reference value to evaluate our GMI model simulation. To compare the GEOS-5

196 assimilated tropospheric partial column above 500 hPa with GMI-CTM ozone

197 simulation, we use the same tropopause as defined by the lower of the 3.5 potential

198 vorticity units (PVU) isosurface and the 380 K isentropic surface.

**3 Results**

**3.1 Temporal and spatial distribution of SHTOM in GMI-CTM and GMAO GEOS-5 assimilated ozone product**

202 Figure 1 shows the spatial pattern of southern hemispheric partial column ozone (from

203 500 hPa to the tropopause) in four seasons averaged over 2005 to 2011 from the GMAO

204 GEOS-5 assimilated dataset and the GMI-CTM hindcast simulations. To account for a

205 low bias in the GEOS-5 ozone product (Wargan et al., 2015), we added 2.5 DU to the

206 assimilated column in the tropics (0-30°S). The GMI-CTM simulation reproduces the

207 seasonality and spatial distribution of southern hemispheric ozone maximum as shown in

208 GEOS-5 assimilated product with a) the elevated ozone centered over the Atlantic Ocean

209 from the equator to 40°S; b) the ozone maximum extending southeastward to southern

210 Africa and the Indian Ocean in the latitude band of 30°S-45°S; c) the relatively weaker

211 enhancement extending westward to South America and the tropical southeastern Pacific.

212 The elevated ozone maximum is strongest in austral winter-spring and weakest in austral

213 fall. Both GMI-CTM and GEOS-5 assimilation show the very low tropospheric ozone

214 over the western Pacific and the tropical eastern Indian Ocean, where the ozone - poor

215 marine boundary layer air is lifted into the upper troposphere (Folkins et al., 2002;

216 Solomon et al., 2005).

**3.2 Subregions of SHTOM**

218 The tropospheric ozone distribution depends on the advection and mixing, their proximity

219 to the polluted area, and descent of ozone-rich air from the stratosphere. We show in

220 Figure 2 the maps of simulated $O_3$ and $StratO_3/O_3$ at 430 hPa averaged over 1992 to 2011



in September, when the southern hemisphere biomass burning peaks. The $StratO_3/O_3$ ratio
represents fraction of tropospheric ozone from stratosphere and is used to identify the
regions with distinct stratospheric input. Differences in the spatial patterns of the
maximum/minimum in ozone mixing ratio and $StratO_3/O_3$ ratio identifies regions where
ozone is affected by factors other than the stratospheric input.
The region with minimum stratospheric ozone contribution occurs along the equator. It
extends southward to approximately 10°S over South America and further south to
approximately 15°S over the Indian Ocean and the Maritime Continents. In the tropics,
the southward extension of regions with minimum stratospheric ozone contribution
shows strong meridional variation that is closely related to the Walker Circulation. In this
tropical meridional circulation air rises over the Maritime Continents (together with deep
convection) and descends over the eastern Pacific (Bjerknes, 1969). Similar meridional
circulation is found over the Atlantic with rising due to radiative heating over tropical
Africa and South America and sinking due to radiative cooling over the tropical Atlantic
(Julian and Chervin, 1978). The longitudinal variation of ozone at 430 hPa in the tropics
is in agreement with the changes of $StratO_3/O_3$, showing ozone minimum over South
America, tropical Africa and Maritime Continents. Within the Atlantic, despite of the
smaller stratospheric contribution, the tropics have higher ozone mixing ratio (>80 ppb)
than the subtropics at 430 hPa, and other sources must also contribute to the ozone
maximum over tropical south Atlantic. Ozone over the tropical southeastern Pacific is
also slightly elevated. The maximum stratospheric influence is found over the Southern
Ocean centered on 30°S, co-located with the tropospheric $O_3$ maximum over these
regions. Both ozone and $StratO_3/O_3$ over the subtropics show strong longitudinal
variations, with the co-located maxima over the south Indian Ocean. The ozone minimum
at 430 hPa at 30°S occurs over the eastern Pacific region, while the minimum
contribution of the stratospheric input is over the south Atlantic region. Given the spatial
variations of the maximum/minimum in $StratO_3/O_3$ ratio and ozone mixing ratio, we
separate the southern hemispheric ozone maximum into four sub-regions: 1) Tropical
southeastern Pacific (0-20°S, 150°W-60°W); 2) Tropical South Atlantic (0-15°S, 60°W-
40°E); 3) Subtropical South Atlantic (15°S-45°S, 60°W-40°E); 4) Subtropical South
Indian Ocean (15°S-45°S, 40°E-150°E). We examine and quantify the relative roles of





dynamics and chemistry on the IAV of tropospheric ozone variations over these selected
regions during the past twenty years.
Figure 3 compares the anomalies of modeled and assimilated tropospheric ozone mixing
ratio at 270 hPa and 430 hPa as well as the anomalies of corresponding upper
tropospheric ozone columns (UTOC, integrated from 500 hPa to the tropopause) over
two tropical sub-regions (tropical south Atlantic and tropical southeastern Pacific) from
2005 to 2011. The anomalies are calculated by removing the monthly mean averaged
from 2005 to 2011. The short time scale variations in the model simulation tend to be
greater compared to that in the assimilated ozone products, especially over the tropical
south Atlantic region. But in general, the GMI-CTM hindcast simulation captures the
assimilated IAV of the tropospheric ozone at these two levels as well as for the UTOC.
Over the tropical south Atlantic, the modeled IAV agrees with the phase changes of
assimilated ozone IAV but the simulation overestimates the assimilated ozone maximum
in 2010 and underestimates the assimilated minima in 2007 and 2011 at both levels. Over
the tropical southeastern Pacific, the IAV is influenced by ENSO relate changes in
dynamics (e.g., Ziemke et al., 2010; Oman et al., 2013; 2011). The simulation matches
much of the assimilated IAV, showing high ozone anomalies after 2005, 2010 La Nina
year and negative ozone anomalies after strong El Niño year in 2009. However, during
October 2006 to January 2007, the simulation shows a pronounced ozone peak, especially
at 270 hPa, which is not seen in the assimilated ozone. Logan et al. (2008) examined
interannual variations of tropospheric ozone profiles in October-December between 2005
and 2006 based on the satellite observations from Tropospheric Emission Spectrometer
(TES). The TES data agrees with what we found in the GMI-CTM model simulation,
showing ozone enhancement over the tropical southeastern Pacific (150°W-60°W, 0-
12°S) region in November 2006 relative to 2005 (~5-10 ppb at 250 hPa and 0-5 ppb at
400 hPa, Figure 3 of Logan et al., 2008). The agreement between TES and GMI-CTM
indicates a possible low bias of GMAO assimilated ozone during late 2006, as a result of
the lack of emissions in the assimilation (Wargan et al., 2015).
Figure 4 shows the similar comparison as Figure 3, but over the two subtropical regions.
Over the South Atlantic region, the assimilated ozone has similar but stronger IAV than
that over the tropical southeastern Pacific region, showing the largest ozone year-by-year





variation (~20ppb at 270 hPa) from October 2009 to October 2010, and the GMI-CTM
simulation reproduces this variation quite well. Over the South Indian region, the
assimilated ozone has weaker IAV than over other regions. Our model reproduces most
of the variations in magnitude and phase, but shows anti-phase variations in late
2006/early 2007, which substantially affected the calculated correlation coefficients
between model and assimilated ozone. The simulated upper tropospheric ozone column
reproduces well the IAV in the assimilated ozone column except for the late 2006. In
general, agreement between the simulated and assimilated results confirms the suitability
of the model for investigations of the controlling factors on the tropospheric ozone IAV
over these regions.
The left column of Figure 5 presents the monthly profiles of correlation coefficients
between the simulated ozone and $StratO_3$ over the four sub-regions. Strong positive
correlations between $StratO_3$ and $O_3$ are observed in most seasons in the upper
troposphere even over two tropical regions. Stratospheric influence plays a big role
during austral winter-spring and reaches its seasonal maximum in August, when the
subtropical jet system is strongest and moves to its northern-most location. Over the two
subtropical regions, the strong stratospheric influence persists throughout the whole
troposphere  (r > 0.8 at 700 hPa) in August. Over tropical south Atlantic region, the
strong stratospheric influence is limited to the upper troposphere in austral winter-spring
and decreases sharply with decreasing altitude. Over the tropical southeastern Pacific, the
strong stratospheric influence persists year-long at the upper troposphere and reaches as
low as ~400 hPa except for December.
The right column of Figure 5 shows the seasonal profiles of correlation coefficients
between ozone and ozone from emissions ($emissO_3$). The $emissO_3$ is the difference
between the simulations with varied and constant emission. Over the two subtropical
regions, there are two seasonal maxima in the correlations between ozone and $emissO_3$.
The first occurs in September at the lower troposphere and decreases with increasing
altitude, the second is in December/January showing opposite vertical gradient with
stronger correlations in the upper and middle troposphere. Over the tropical southeastern
Pacific region, the influence from emissions shows a similar double-peak pattern, but
with the first maximum localized at the surface and the second peak localized in the





upper troposphere. Over the tropical south Atlantic, the influence of emissions is very
small. South America and southern Africa are two major nearby burning regions.
Emissions over South America have much larger IAV than those over southern Africa,
although Africa emission is larger in absolute terms (Sauvage et al., 2007; Liu et al.,
2010; Voulgarakis et al., 2015). Sauvage et al (2007) argued that emissions over the
eastern regions (India, South-East Asia, Australia) could be transported southward in the
upper troposphere through the Tropical Easterly Jet and affect ozone over Africa, the
Atlantic and Indian Ocean (Hoskins and Rodwell, 1995; Rodwell and Hoskins, 2001).
Meanwhile, emissions over the eastern region also show large IAV (Voulgarakis et al.,
2015). Therefore, the interannual emission changes in South America (0-20°S, 72.5°W-
37.5°W), southern Africa (5°S-20°S, 12°E-38°E) and the eastern region (70°E-125°E,
10°S-40°N) may all affect the IAV of ozone due to emission changes in the southern
hemisphere. In this study, we rely on tagged CO simulation to quantify the influence of
biomass burning emissions from these three burning regions during months when
emission IAV contributes significantly to the IAV of ozone.
In the next section, we choose August (the seasonal maximum of stratospheric input into
the lower troposphere), September and December (the seasonal maximum of emission
contribution) as three example months to examine the relative roles of different factors on
IAV of tropospheric ozone over these regions.
**3.3 Factors controlling IAV in ozone in the middle and upper troposphere**
**3.3.1 South Atlantic Region**
Figure 6 shows the multiple regression results over the South Atlantic region. It compares
the simulated ozone anomalies to that calculated from two regression variables: $StratO_3$
and $EmissO_3$ at 270 hPa and 430 hPa in August, September and December. The fitted
ozone anomalies in generally reproduce the IAV obtained from the GMI-CTM
simulation. The explained proportion of variability in simulated ozone anomalies by
$StratO_3$ and $EmissO_3$ is mostly above 50% and reaches as high as ~ 76% in December, at
270 hPa, which demonstrates that $StratO_3$ and $EmissO_3$ are sufficient to explain the IAV
of tropospheric ozone over the south Atlantic region. In August at 430 hPa, the fitted





ozone anomalies have a slightly weaker correlation with the simulated ozone and show
less IAV compared to the ozone anomalies in GMI-CTM.
Figure 7 exhibits regression results in a way that highlights the relative contributions of
the IAV of stratospheric input and emission on the IAV of ozone over South Atlantic.
The three panels represent results from August, September and December from 1992 to
2011. Each panel has two columns, which illustrate the respective contribution from
changes in $StratO_3$ and $EmissO_3$ on the IAV of ozone mixing ratio. The left column of
each panel compares the anomalies of $StratO_3$ (blue) and simulated ozone mixing ratio
(black) from the GMI-CTM model at 270 and 430 hPa. The right column compares the
simulated $O_3$ residual after removing the regression from $StratO_3$ (black line) and
$EmissO_3$ (green line) at these two levels. The regression coefficient (β) and its 95%
confidence level are labeled in each panel and help us to determine whether the
corresponding contribution is significant to explain the variation of simulated ozone. As
discussed before, $EmissO_3$ reflects the effects from surface emission changes on ozone
variations at interannual time scale. The stratospheric input reaches its seasonal
maximum in August, during which the stratospheric contribution is significant
throughout the troposphere, explaining about 70% of the simulated ozone variance at 270
hPa and 41% at 430 hPa. The contributions from emission changes are very small and
insignificant at these two levels in August. In September, the IAV of stratospheric input
explains about 53% of the IAV in ozone at 270 hPa. The contribution decreases but is
still significant at 430 hPa. The IAV of surface emissions contributes substantially to the
IAV of ozone in September. The influence of emissions exceeds that of the stratosphere
and explains about 50% of IAV in ozone at 430hPa. In December, the contributions from
stratospheric input on the IAV of ozone are dominant (~60%) at 270 hPa but insignificant
at 430 hPa. Emission influence is significant at both levels. However, unlike that of
September, the influence of emissions on IAV of tropospheric ozone is great at 270 hPa
(~40%) than at 430 hPa (~36%). We quantify emission contributions from three burning
regions using a tagged CO simulation. Figure 8 shows standardized anomalies of the
tagged CO tracers over South Atlantic from three burning source regions, including
southern Africa (red), South America (blue) and eastern region (green) and their
comparison with the $EmissO_3$ at 270 and 430 hPa in September and December from 1992



to 2011. The direct downwind transport of emissions from South America contributes
most to the ozone variability from emissions over this region in September at both levels
and the effects are most significant in the lower level (~58% at 430 hPa). In the upper
troposphere, besides the contribution from S. America, the uplift and cross-equator
transport of pollutants from eastern region also contributes (>10%) to the ozone variation
over South Atlantic region. The contribution from southern Africa is small and less than
10% at both levels. We also note that both $StratO_3$ and $emissO_3$ show a minimum in 2009
and a maximum in 2010. There was a strong El Niño event in the year 2009/2010. Neu et
al. (2014) identified the increased stratospheric circulation in 2010 driven by El
Niño/easterly QBO based on TES data. A few other studies (e.g., Chen et al., 2011;
Lewis et al., 2011) found that combined effects of 2009/2010 El Niño and warmer than
normal Atlantic SST produced a severe drought over S. America and caused extensive
biomass burning emission in 2010 dry season. Therefore, the agreements between
changes in the $StratO_3$ and $emissO_3$ over 2009/2010 are at least partly driven by ENSO.
Similar tropospheric ozone anomalies are observed after 1997 and 2006 El Niño event.
Olsen et al. (2016) examined the magnitude and spatial distribution of ENSO effects on
tropospheric column ozone using the assimilated fields and found a statistically
significant negative response of tropospheric column ozone to ENSO over South Atlantic
Ocean.
In December, emissions from South America and southern Africa do not contribute
substantially to the IAV of $emissO_3$. Emissions from eastern region dominate, explaining
83% and 77% variance of $emissO_3$ IAV at 270 hPa and 430 hPa. The eastern pollutants
have the strongest influence at the upper troposphere because of their transport pathway
as discussed in Sauvage et al. (2007). Therefore, the emission contribution of
tropospheric ozone IAV in December shows an opposite vertical structure to that seen in
September.
In summary, over the South Atlantic region, the stratospheric input plays a dominant role
in the upper troposphere with a seasonal maximum in August. At 430 hPa the
contribution from emission changes to the IAV of ozone exceeds that of stratospheric
input in September and December. A tagged CO simulation from 1992 to 2011 shows the
direct downwind transport of pollutants from South America is the largest contributor to



emissO$_3$ in September, and it is strongest near the surface. In December, cross-equator
transport of eastern region pollutants is the most important source of IAV due to
emissions, and the effects are strongest in the upper troposphere.

### 408  3.3.2 South Indian Ocean

Over the south Indian Ocean, the fitted and simulated ozone anomalies are in excellent
agreement (Figure 9). The explained proportion of variability in simulated ozone
anomalies by StratO$_3$ and EmissO$_3$ is as high as ~ 88% in August at 270 hPa. We show
relative contribution to the IAV in ozone due to stratospheric input and emission as
obtained from multiple linear regression in Figure 10. In August and September,
stratospheric input contributes more than 85% to ozone IAV at 270 hPa. The
stratospheric contribution decreases slightly but is still dominant and significant at 430
hPa (~50% in August and 64% in September). The emission contribution, which is
mainly from downwind transport of pollutants from S. America and southern Africa
(Figure 11), is most important at 430 hPa in September but accounts for only 27% of
ozone IAV. The emission contribution is smaller in August. In December, both
stratospheric input and surface emission influence the IAV of ozone. The contribution
from stratospheric input slightly exceeds that from emissions at 270 hPa and becomes
slightly weaker at 430 hPa. Examining the tagged sources simulation shows that emission
from eastern regions is the largest sources of ozone IAV at 270 hPa and 430 hPa in
December with a stronger influence at the upper troposphere (Figure 11).
These results show that stratospheric ozone makes a significant contribution to the
tropospheric ozone variability over the South Indian Ocean, with the largest influence in
the upper troposphere in austral winter. Emission influence from nearby pollution in the
boundary layer is relatively weak and only significant in September, one month after the
southern hemisphere peak-burning season. In the upper troposphere, the cross-equator
transport of pollutants from the eastern region is the major emission source affecting the
ozone variability. The influence peaks in December at the upper troposphere and extends
to the middle troposphere.

### 433  3.3.3 Tropical South Atlantic





In the upper troposphere, lightning produces nitrogen oxides ($NO_X$) and promotes the
photochemical ozone production (e.g., Pickering et al., 1993). Murray et al. (2013) shows
that the IAV of tropical tropospheric ozone column is sensitive to the IAV of lightning
over the tropical south Atlantic region. We therefore add the lightning $NO_X$ as the third
variable besides $StratO_3$ and $EmissO_3$. We test whether the addition of lightning $NO_X$
improves the regression model significantly. Figure 12 shows the comparison between
simulated and fitted ozone anomalies without and with lightning $NO_X$. During the "dry
season" months of August and September, when the subtropical jet related STE (Karoly
et al., 1998; Bals-Elsholz et al., 2001; Nakamura and Shimpo, 2004) reaches a seasonal
maximum, the lightning activities reach a seasonal minimum over the southern
hemisphere. The fitted ozone anomalies based solely on $StratO_3$ and $EmissO_3$ (red) show
high correlations (r = 0.8 in August, r = 0.74 in September) with that simulated from
GMI-CTM at 270 hPa. Agreement between simulated and fitted ozone does not change in
August and improves slightly in September by adding lightning $NO_X$ in regression. In
September, the simulated ozone anomaly shows a minimum (~ -6 ppb) in 2007 and a
peak (~ 5ppb) in 2010 at 430 hPa, but the IAV from 2007 to 2010 is almost missing in
the fitted ozone anomaly, which indicates that other factors drive the IAV of ozone over
tropical south Atlantic during this period. During the "wet season" month of December,
the lightning activity reaches its seasonal maximum. Our regression based on $StratO_3$ and
$EmissO_3$ does not capture well the IAV of GMI-CTM simulated ozone at either level.
The fitted ozone reproduces many of the IAV of simulated ozone after including
lightning $NO_X$ in the regression, indicating a strong influence from the lightning NOX in
December.
Figure 13 shows the regression results of relative contributions of stratospheric input and
surface emission on the IAV of ozone. As discussed above, the tropical south Atlantic is
in the descending branch of the Walker Circulation. Therefore, even though this region is
located in the tropics, the IAV of stratospheric input still plays a dominant role and
explains 64% in August and 50% in September of ozone variance in the upper
troposphere. The stratospheric contribution, associated with radiative descent over this
region, drops to less than 30% at 430 hPa but is still significant during these two months.
Emission influences are not significant at either level in September. Examination of the



simulation shows that emission contribution is limited even at lower levels; the emission
contribution becomes significant and explains ~30% variance of ozone at ~700 hPa (not
shown). In December, neither stratospheric input nor emission contributes much to the
IAV of ozone.
In the model, the lightning emissions take place in connection with deep convective
events (Allen et al., 2010). Increase in deep convection produces more upper tropospheric
$NO_X$ from lightning, which results in more ozone production. On the other hand, deep
convection affects the upper tropospheric ozone budget through its direct transport of
surface air. In December, biomass burning in the Southern Hemisphere is at its seasonal
minimum. Air over tropical south Atlantic is relatively clean with low CO (Liu et al.,
2010). Deep convection over a clean region reduces upper tropospheric ozone due to
mixing. This effect could be positive if deep convection happens over a polluted region
with relatively high ozone and its precursors (Lawrence et al., 2003). Use of the
correlation to identify influence from the lightning $NO_X$ does not separate the two
outcomes of IAV in convection, thus the sign of the correlation between variations in
lightning $NO_X$ and upper tropospheric ozone can be positive or negative. The correlation
is positive if the contribution from lightning $NO_X$ exceeds the contribution from
convective transport or if transport of polluted air increases ozone. The correlation is
negative if transport of clean air overwhelms ozone production from lightning $NO_X$.
Figure 14 compares the model residual after removing the contributions from $StratO_3$ and
$EmissO_3$ with the lightning $NO_X$ at 270 hPa in September and December. In September
the IAV of lightning plays a minor but significant role in the IAV of ozone in the upper
troposphere. In December, the changes in lightning $NO_X$ have a significant impact on the
ozone IAV, but show a negative correlation, which indicates that the transport and
mixing of clean surface air exceeds ozone production from lightning $NO_X$ emissions with
a net negative impact of IAV in convection.
**3.3.4 Tropical southeastern Pacific**
Figure 15, 16 and 17 show the similar comparisons but over the tropical southeastern
Pacific region. The fitted ozone anomalies show moderate but still significant correlations
with that simulated from GMI-CTM in August and September. In December, the fitted



ozone IAV agrees very well with the GMI-CTM simulated ozone IAV at 270 hPa. At 430
hPa the agreement collapses and the fitted ozone does not show strong IAV as seen in the
GMI-CTM simulated ozone (Figure 15). Figure 16 shows that IAV in stratospheric input
significantly affects the ozone IAV during these three months, explaining 25-38% of the
variance of simulated ozone at 270 hPa. Emissions contribution is quite small in August
and September, but is significant and explains 28% of simulated ozone IAV in December
at 270 hPa. The tagged CO simulations show that the tropical southeastern Pacific region
is influenced by nearby pollutants from South America, and also by the cross-equator
transport of pollutants from the eastern region (Figure 17). Previous studies (e.g.,
Chandra et al., 1998; 2002; 2009; Sudo and Takahashi, 2001; Ziemke and Chandra, 2003;
Doherty et al., 2006; Oman et al., 2011) show that ENSO has its strongest impact in the
tropical Pacific basin. In August, the ITCZ is located at its northernmost location north of
the Equator. Radiative sinking motion still dominates over the tropical southeastern
Pacific in the middle - upper troposphere (Liu et al., 2010). Therefore, the emissions
contribution from South America is quite small at 430 hPa and 270 hPa as shown in
Figure 16. During an El Niño year, warmer SST with increase convection and large-scale
upwelling start occurring in August, inhibiting the radiative sinking motion and resulting
in ozone decrease in the middle-upper troposphere over this region. Our comparison
shows strong negative correlation in August between IAV of middle-upper tropospheric
ozone anomalies over this region and Niño 3.4 index during the past twenty years (Figure

515  18).

**4 Summary and Discussion**
Both model simulations and GEOS-5 assimilated ozone product derived from OMI/MLS
show a tropospheric ozone column maximum centered over the south Atlantic from the
equator to 30S. This ozone maximum extends westward to South America and the eastern
equatorial Pacific; it extends southeastward to southern Africa and the south Indian
Ocean. In this study, we use hindcast simulations from the GMI model of tropospheric
and stratospheric chemistry, driven by assimilated MERRA meteorological fields, to
interpret and quantify the relative importance of the stratospheric input and surface



emission to the interannual variations of tropospheric ozone over four sub-regions of the
SHTOM from 1992 to 2011. Over the SHTOM region, IAV in the stratospheric
contribution is found to be the most important factor driving the IAV of ozone, especially
over the upper troposphere, where $O_3$ changes have strong radiative effects (Lacis et al.,
1990). The IAV of the stratospheric contribution explains a large portion of variance in
the tropospheric ozone especially during the austral winter season, even over two selected
tropical regions. The strong influence of emission on ozone IAV is largely confined to
the South Atlantic region in September.
Although the SHTOM looks like a continuous feature in the southern hemisphere, our
study shows that the relative importance between stratospheric input and surface
emissions changes over different subregions at different altitude. Over the two extra-
tropics regions, the IAV of stratospheric contribution explains at least 50% of variance of
the tropospheric ozone during its winter season. The IAV of ozone over the south Indian
Ocean is dominantly driven by the IAV of stratospheric ozone contribution (>64%) with
little or no influence from surface emissions at 270 hPa and 430 hPa. Over the south
Atlantic region, besides the stratospheric ozone input, the IAV of surface emissions from
South America and southern Africa also play a big role on the IAV of ozone, especially
in the lower levels. The influence from emission exceeds that from the stratospheric
contribution on the ozone variability in September at 430 hPa. In December, the emission
influence mainly from remote transport of pollutants from eastern region is relatively
high in the upper troposphere and decreases with the decreasing altitude.
Compared to the extra-tropics regions, the two tropical regions have a smaller influence
from stratospheric input but the influence is still significant at both 270 hPa and 430 hPa
in August and September. Over tropical south Atlantic region, the IAV of stratospheric
input plays a dominant role and explains 64% in August and 52% in September of the
ozone IAV at 270 hPa. The stratospheric contribution drops to less than half of that at
270 hPa but is still significant at 430 hPa. Emission contributions are not significant at
these two levels, even during September. Our model shows that the IAV of ozone is
partially driven by the IAV of lightning in September. In December, the changes in
lightning $NO_X$ have a significant impact on the ozone IAV, but show a negative
correlation, which indicates that the transport and mixing of clean surface air exceeds





ozone production from lightning $NO_X$ emissions with a net negative impact of IAV in
convection. Over the tropical southeastern Pacific, IAV in stratospheric input
significantly affects the ozone IAV during these three months, explaining 25-38% of the
variance of simulated ozone at 270 hPa. Emissions have little or no influence in August,
September at 270 hPa and 430 hPa, but are significant in December at 270 hPa,
explaining 28% of simulated ozone IAV. A further comparison of ozone and ENSO
index shows that ENSO, which affects the tropical convection and large-scale upwelling,
shows a strong negative correlation with the IAV of tropospheric ozone over this region.
Therefore, the model simulations/predictions with different convective parameterizations
exhibit large uncertainties over this region as observed in Stevenson et al. (2006).
In this study, our regional analysis based on the GMI-CTM model provides valuable
conclusions on drivers of interannual variability over different subregions of the SHTOM
and how they vary with the altitude. The quantification of their relative contributions on
interannual time scales enhances our understanding of the IAV and, potentially, long-
term trends in the tropospheric ozone and furthermore their effects to the radiative
forcing change in climate.
**Acknowledgement**
All model output used for this article can be obtained by contacting J. Liu (email:
junhua.liu@nasa.gov). I gratefully acknowledge the financial support by NASA's
Atmospheric Chemistry Modeling and Analysis Program (ACMAP) (grants
NNH12ZDA001N). Work was performed under contract with NASA at Goddard. I
would like to thank K. Pickering, L. Oman, A. Thompson, H. Liu for their helpful
discussion.

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



**Figures:**

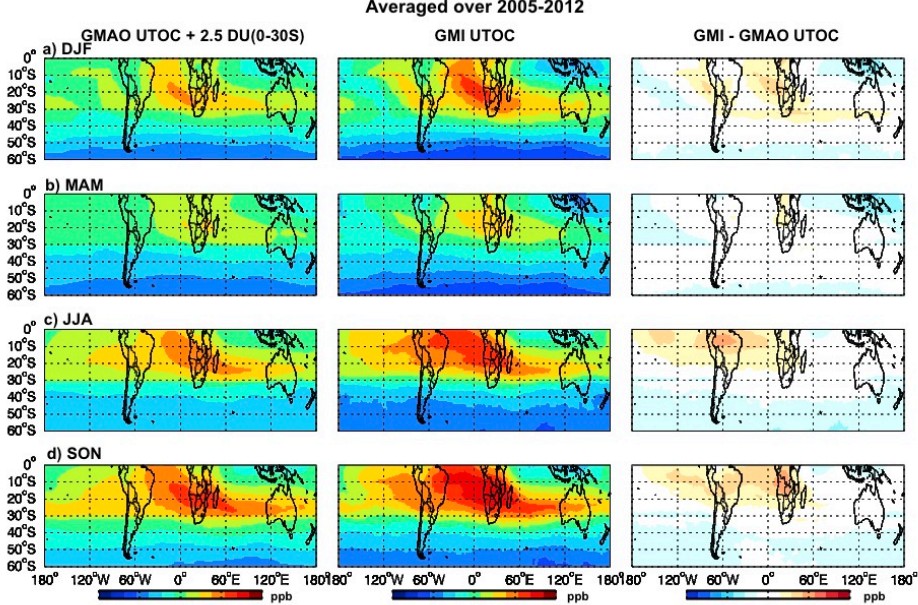


Figure 1: Seasonal climatology of upper tropospheric column ozone (UTOC) (in Dobson Units) for (a)
December-January-February (DJF), (b) March-April-May (MAM), (c) June-July-August (JJA), and (d)
September-October-November (SON) averaged from 2005 to 2012 for GMAO assimilated ozone (left) and GMI-
CTM Hindcast-VE ozone (middle) and their absolute difference (right). The GMAO assimilated ozone has been
adjusted by adding 2.5 DU in 0-30° S based on Wargan et al. (2015).



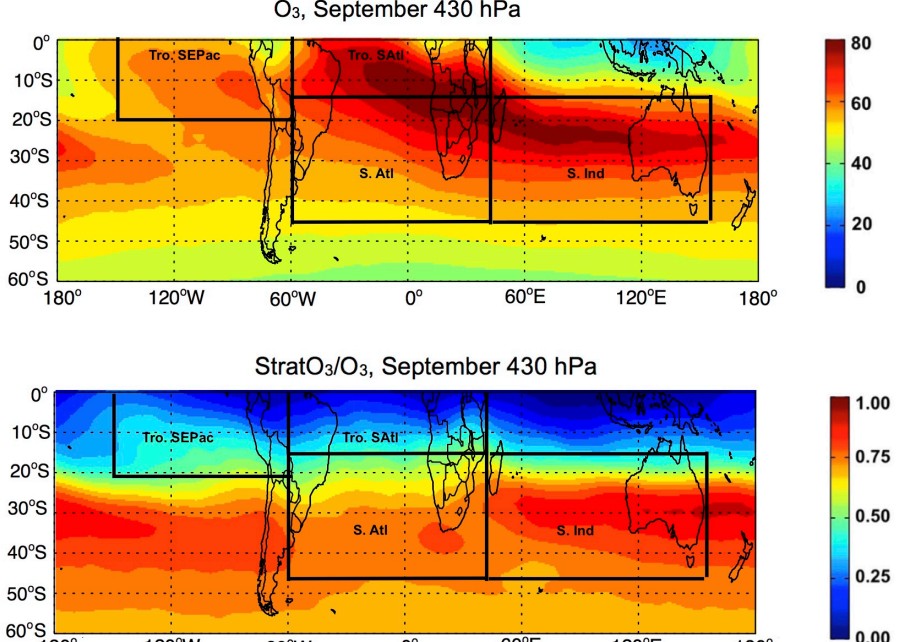

**Figure 2: The simulated ozone (top) and the StratO$_3$/O$_3$ (bottom) at 430 hPa averaged over 1991-2011 in September. Stronger stratospheric influence happens over southern hemisphere centered on 30° S, co-locating with subtropical jet stream regions with descending stratospheric air. The black boxes show four regions discussed in this study. From left to right: (1) Tropical southeastern Pacific (0-20° S, 150° W-60° W); (2) Tropical South Atlantic region (0-15° S, 60° W-40° E); (3) Subtropical South Atlantic region (15° S-45° S, 60° W-40° E); (4) Subtropical South Indian Ocean (15° S-45° S, 40° E-150° E).**

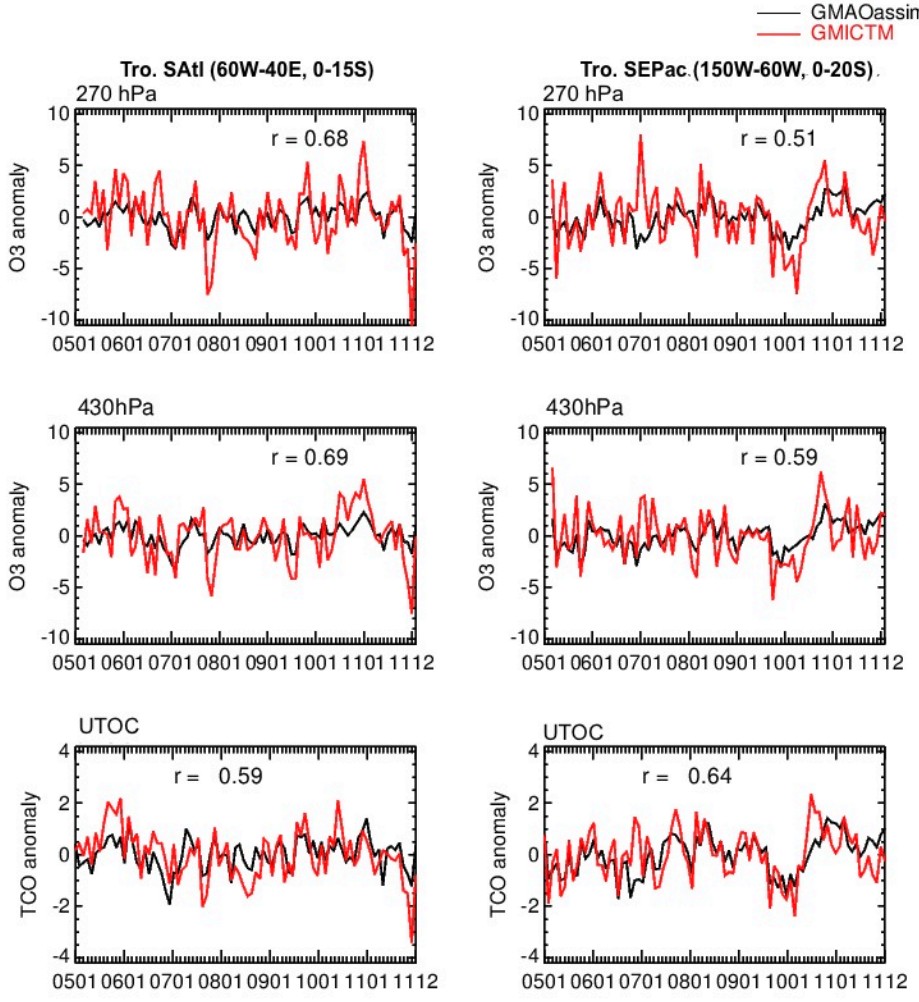

837

**Figure 3: Time series plots of tropospheric ozone anomalies (unit: ppb) from GMAO assimilated data (black)**

**and GMI-CTM (red) at 270 hPa and 430 hPa and upper tropospheric ozone column (UTOC, integrated from**

**500 hPa to the tropopause) anomalies over (left) Tropical South Atlantic region (0-15° S, 60° W-40° E); (right)**

**Tropical southeastern Pacific (0-20° S, 150° W-60° W) from 2005 to 2011.**



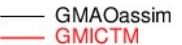

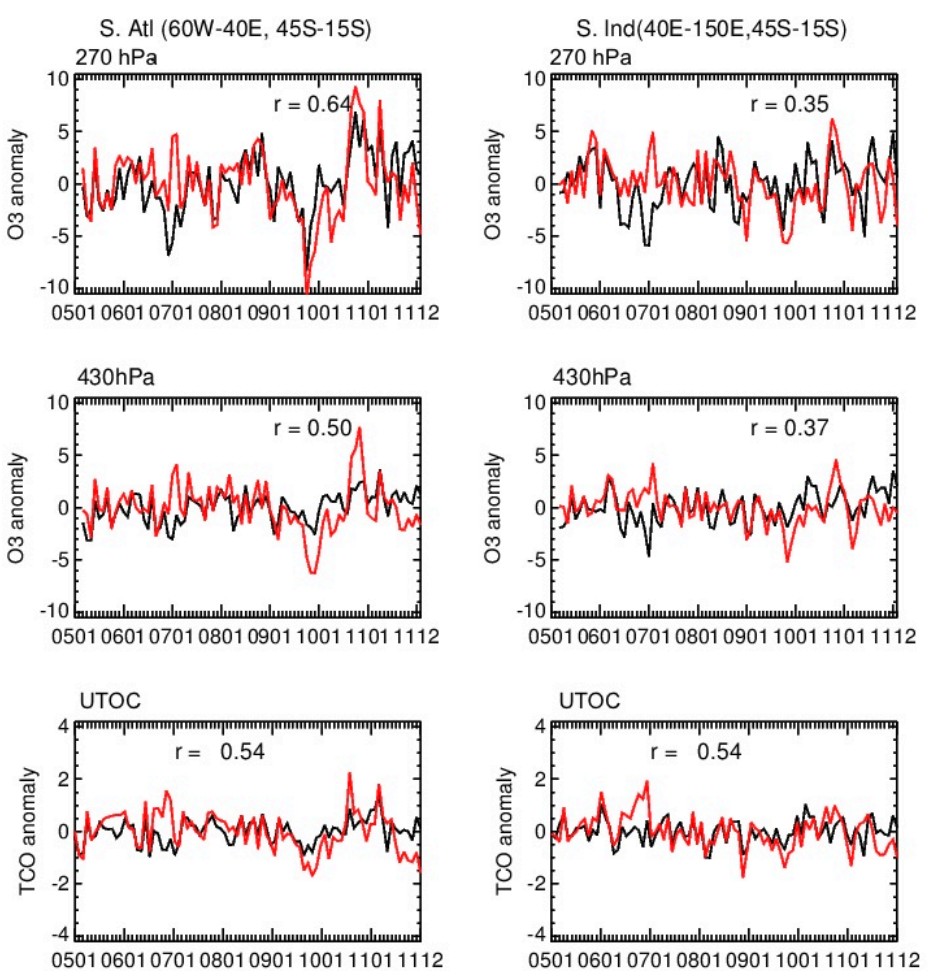

**Figure 4: Time series plots of tropospheric ozone anomalies (unit: ppb) from GMAO assimilated data (black) and GMI-CTM (red) at 270 hPa and 430 hPa and upper tropospheric ozone column (UTOC, integrated from 500 hPa to the tropopause) anomalies over (left) South Atlantic (15° S-45° S, 60° W-40° E); (right) South Indian Ocean (15° S-45° S, 40° E-150° E) from 2005 to 2011.**



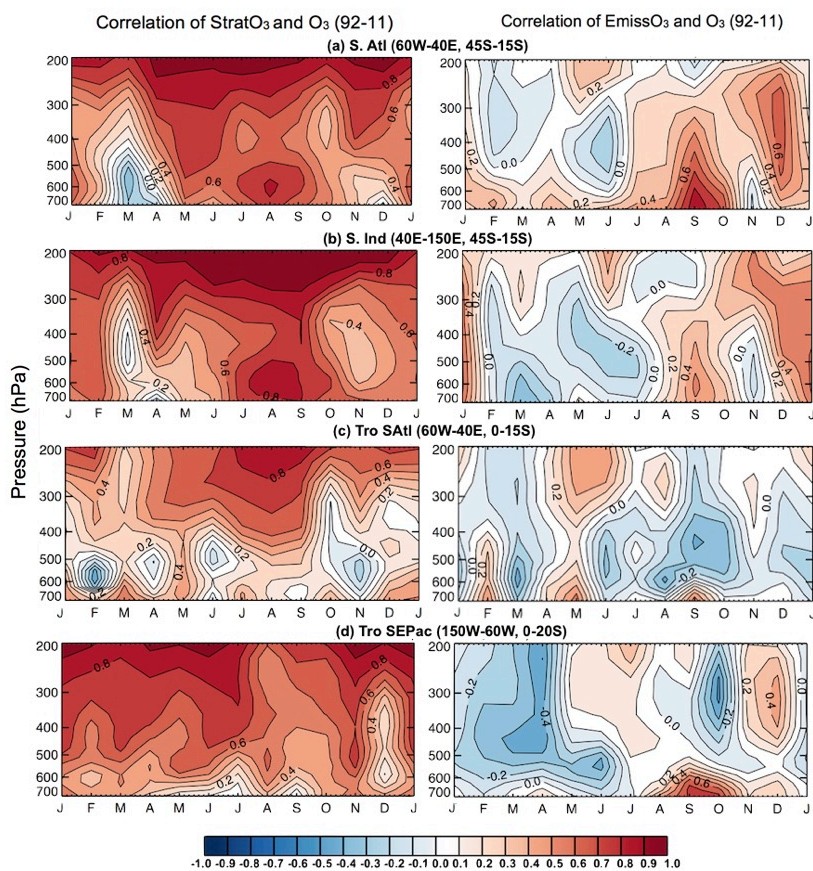


**Figure 5: Monthly profile maps of correlations coefficients between ozone and left) StratO$_3$, right) EmissO$_3$**

**from 1992 to 2011 over (a) South Atlantic (15° S-45° S, 60° W-40° E); (b) South Indian Ocean (15° S-45° S, 40°**

**E-150° E); c) Tropical South Atlantic region (0-15° S, 60° W-40° E); d) Tropical southeastern Pacific (0-20° S,**

**150° W-60° W). Y-axis is pressure in unit hPa.**





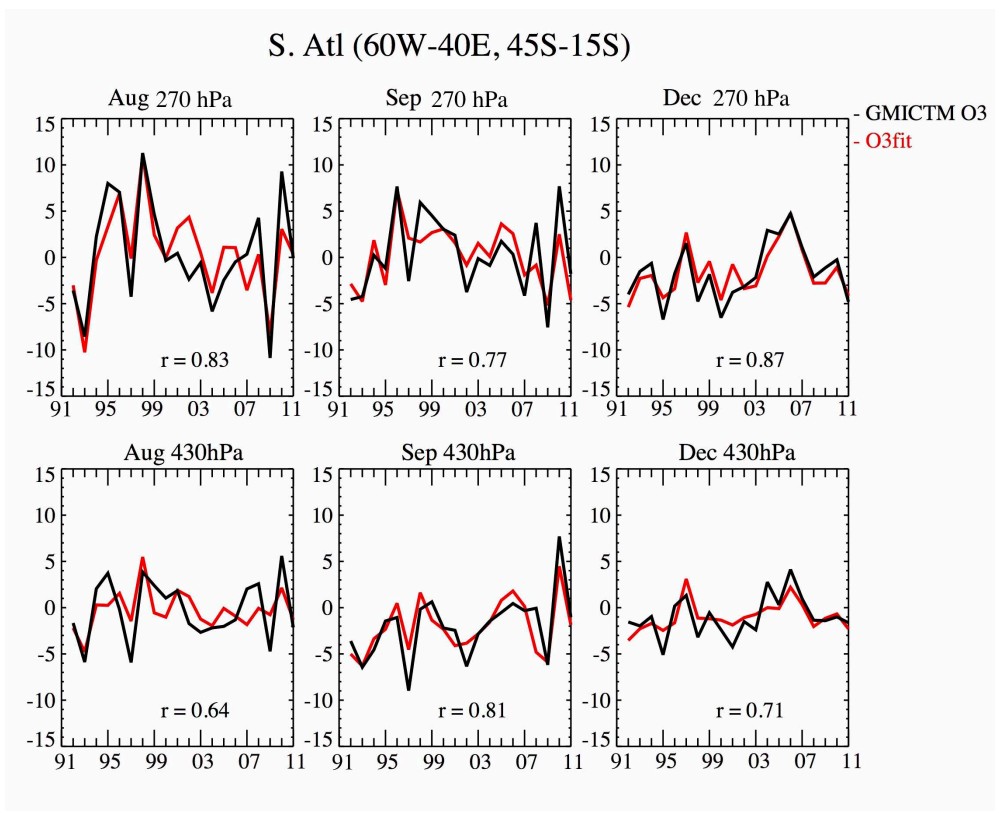


**Figure 6: Comparison of the simulated ozone anomalies and the calculated ozone anomalies relying on two predictor variables: StratO₃, EmissO₃ at 270 hPa and 430 hPa over South Atlantic region. Three panels show results from August (left), September (middle) and December (right) from 1992 to 2011. Unit for y-axis is ppb.**





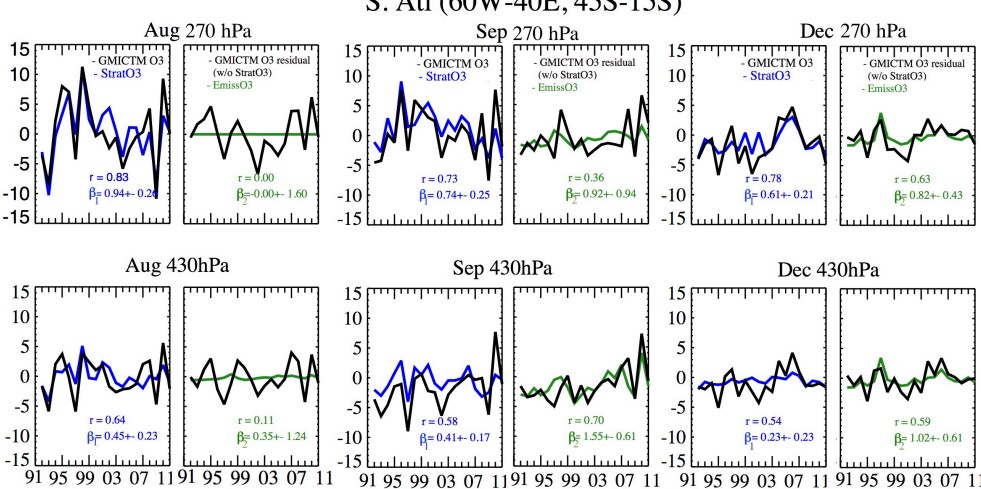


**Figure 7: The multi-regression results of simulated ozone anomalies over South Atlantic region relying on two**
**predictor variables: StratO$_3$ (blue), EmissO$_3$ (green) at 270 hPa and 430 hPa. Three panels show results from**
**August (left), September (middle) and December (right) from 1992 to 2011. Each panel contains two columns.**
**The left column of each panel compares the anomalies of StratO$_3$ (blue) and simulated ozone mixing ratio**
**(black) from the GMI-CTM model at 270 and 430 hPa. The right column compares the simulated O$_3$ residual**
**after removing the regression from StratO$_3$ (black line) and EmissO$_3$ (green line) at these two levels. EmissO$_3$ is**
**calculated from the difference of simulated ozone between the run with yearly-varied emission and the run with**
**constant emission. Unit for y-axis is ppb. The correlation (r), regression coefficient (β) and its 95% confidence**
**level are labeled in each panel.**






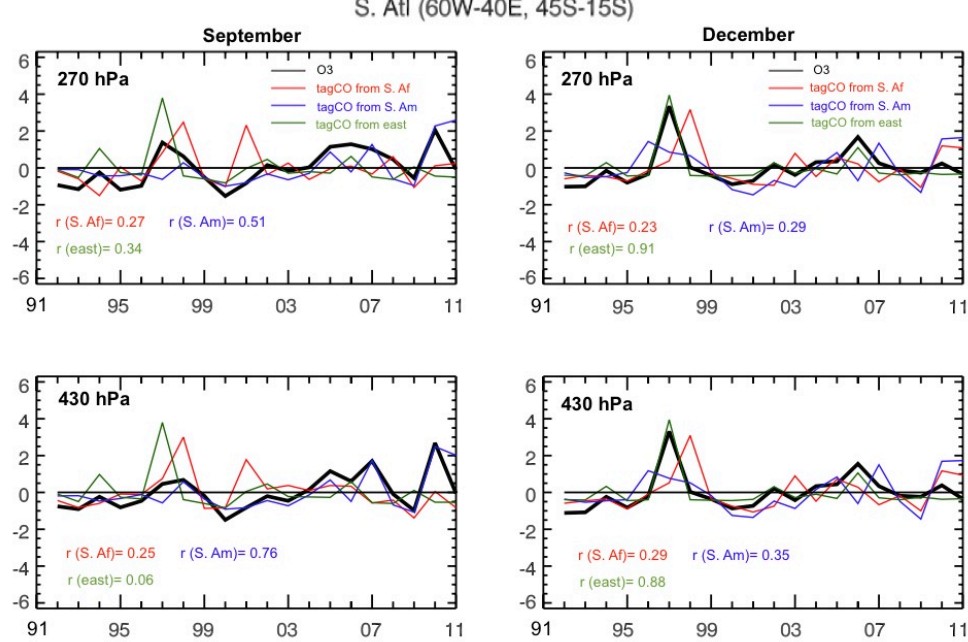


**Figure 8: The standardized anomalies of the tagged CO tracers over South Atlantic from three burning source**
**regions, including southern Africa (red), South America (blue) and eastern region (green) and their comparison**
**with the EmissO$_3$ (black) at 270 and 430 hPa in September and December from 1992 to 2011.**





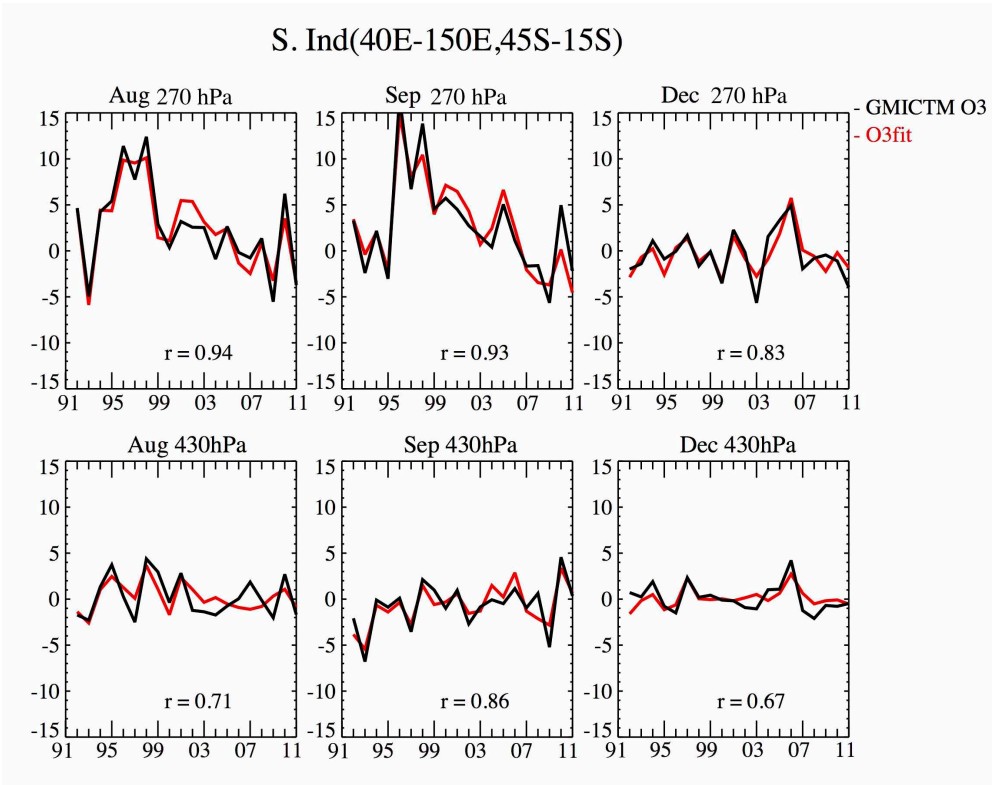


**Figure 9: Comparison of the simulated ozone anomalies and the reconstructed ozone anomalies relying on two**

**predictor variables: StratO$_3$, EmissO$_3$ at 270 hPa and 430 hPa over South Indian Ocean region. Three panels**

**show results from August (left), September (middle) and December (right) from 1992 to 2011. Unit for y-axis is**

**ppb.**



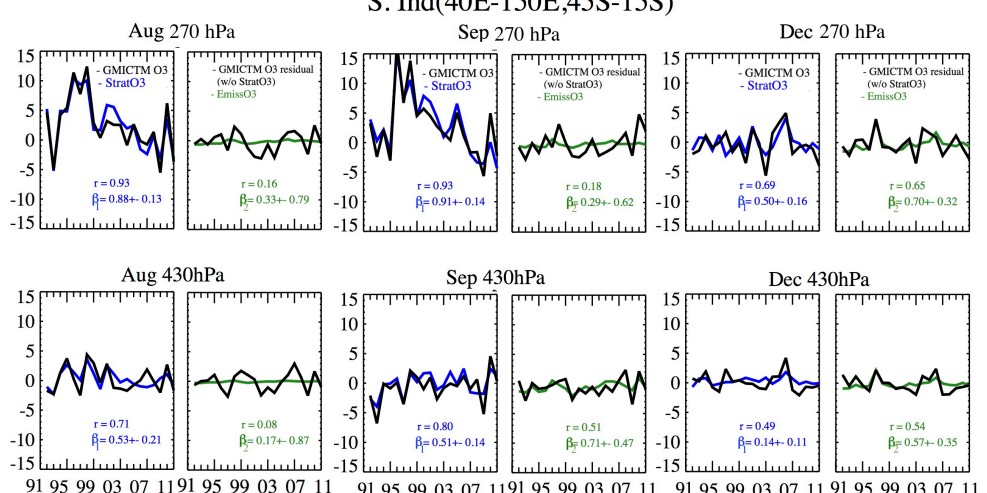

Figure 10: The multi-regression results of simulated ozone anomalies over South Indian Ocean region relying on two predictor variables: StratO₃ (blue), EmissO₃ (green) at 270 hPa and 430 hPa. Three panels show results from August (left), September (middle) and December (right) from 1992 to 2011. Each panel contains two columns. The left column of each panel compares the anomalies of StratO₃ (blue) and simulated ozone mixing ratio (black) from the GMI-CTM model at 270 and 430 hPa. The right column compares the simulated O₃ residual after removing the regression from StratO₃ (black line) and EmissO₃ (green line) at these two levels. EmissO₃ is calculated from the difference of simulated ozone between the run with yearly-varied emission and the run with constant emission. Unit for y-axis is ppb. The correlation (r), regression coefficient (β) and its 95% confidence level are labeled in each panel.





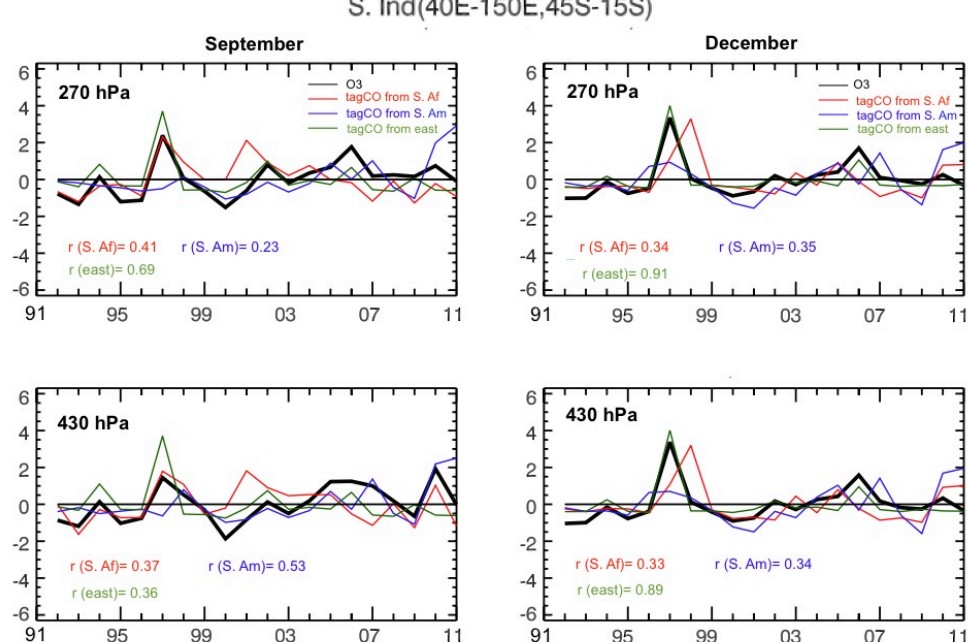

**Figure 11: The standardized anomalies of the tagged CO tracers over South Indian Ocean region from three**
**burning source regions, including southern Africa (red), South America (blue) and Eastern region (green) and**
**their comparison with the EmissO$_3$ (black) at 270 and 430 hPa in September and December from 1992 to 2011.**



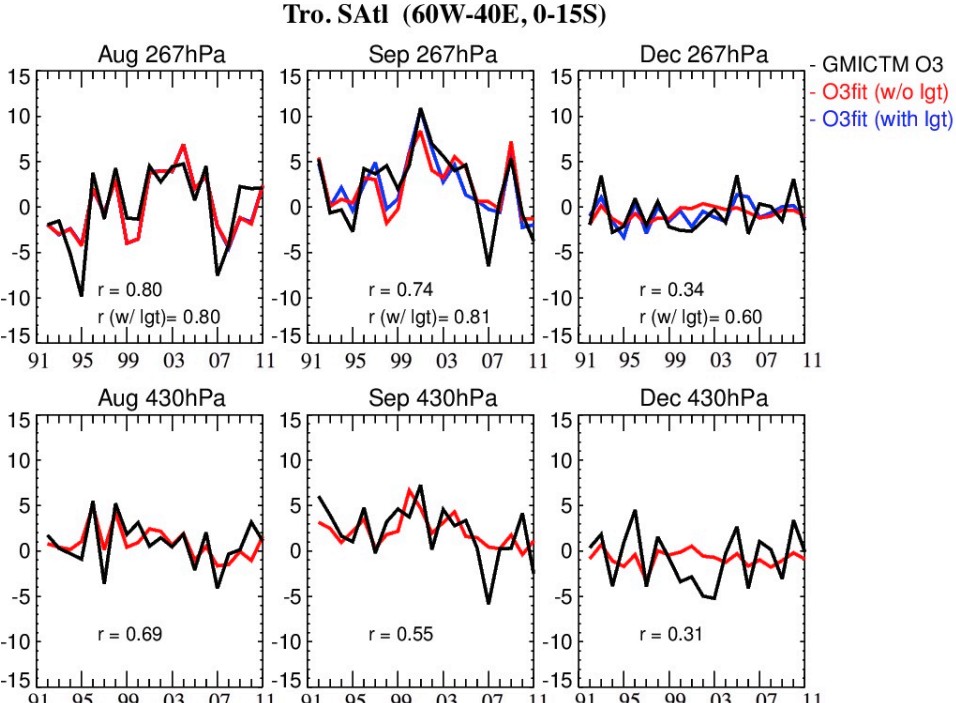


**Figure 12: Comparison of the simulated ozone anomalies and the reconstructed ozone anomalies relying on two**

**predictor variables: StratO$_3$, EmissO$_3$ (red) over Tropical South Atlantic region at 270 hPa and 430 hPa. At 270**

**hPa, the reconstructed ozone anomalies from three predictor variables including lightning NO$_X$ (blue) are**

**added. Three panels show results from August (left), September (middle) and December (right) from 1992 to**

**2011. Unit for y-axis is ppb.**





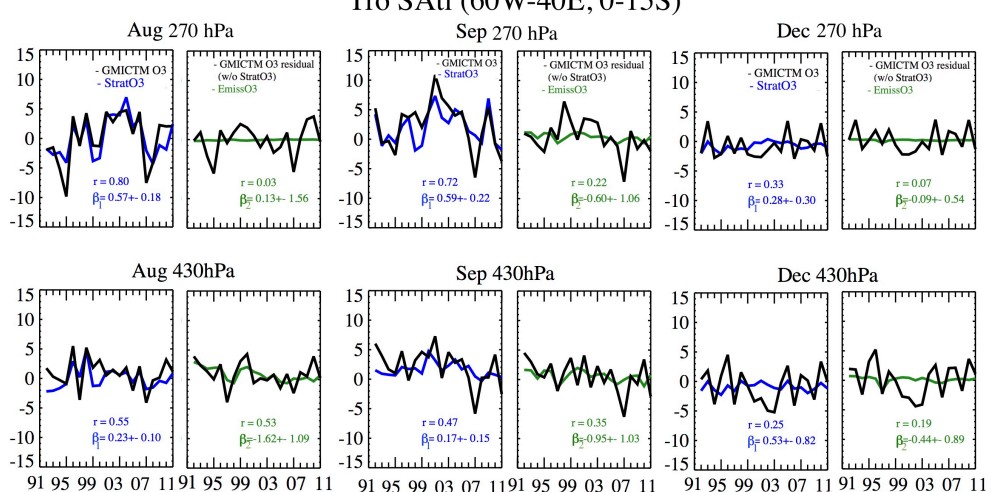


Figure 13: The multi-regression results of simulated ozone anomalies over tropical South Atlantic region relying
on StratO$_3$ (blue), EmissO$_3$ (green) at 270 hPa and 430 hPa. Three panels show results from August (left),
September (middle) and December (right) from 1992 to 2011. Each panel contains two columns. The left column
of each panel compares the anomalies of StratO$_3$ (blue) and simulated ozone mixing ratio (black) from the GMI-
CTM model at 270 and 430 hPa. The right column compares the simulated O$_3$ residual after removing the
regression from StratO$_3$ (black line) and EmissO$_3$ (green line) at these two levels. EmissO$_3$ is calculated from the
difference of simulated ozone between the run with yearly-varied emission and the run with constant emission.
Unit for y-axis is ppb.  The correlation (r), regression coefficient (β) and its 95% confidence level are labeled in
each panel






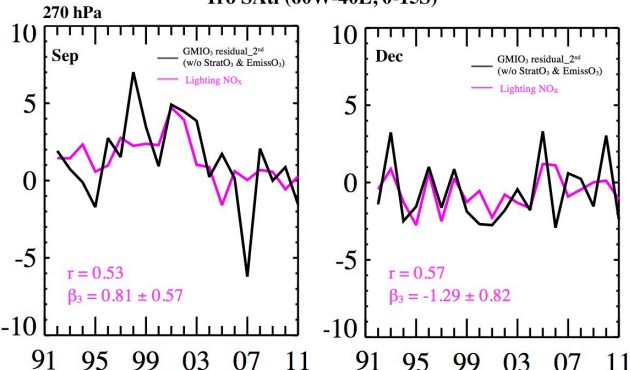

**Figure 14: The comparison between regression of lightning NO_X (magenta) and the ozone residual after**
**removing the regression of StratO3 and EmissO_3 (black) at 270 hPa in September (left) and December (right)**
**over tropical South Atlantic region. The correlation (r), regression coefficient (β) and its 95% confidence level**
**are labeled in each panel.**



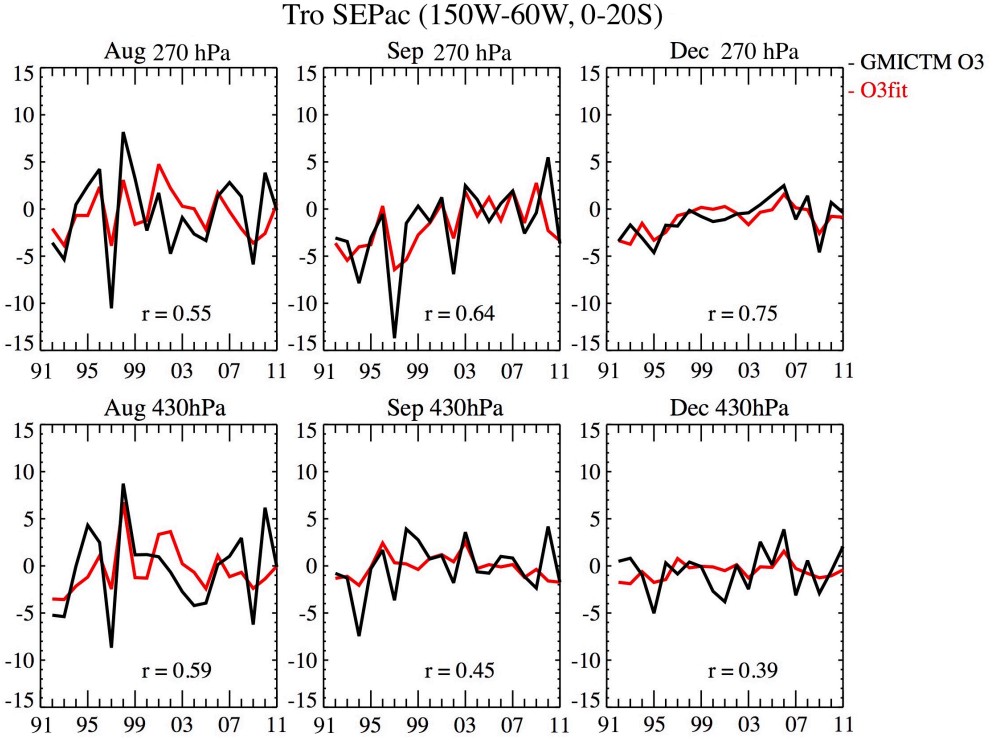


**Figure 15: Comparison of the simulated ozone anomalies and the reconstructed ozone anomalies relying on two**
**predictor variables: StratO$_3$, EmissO$_3$ at 270 hPa and 430 hPa over Tropical southeastern Pacific. Three panels**
**show results from August (left), September (middle) and December (right) from 1992 to 2011. Unit for y-axis is**
**ppb.**



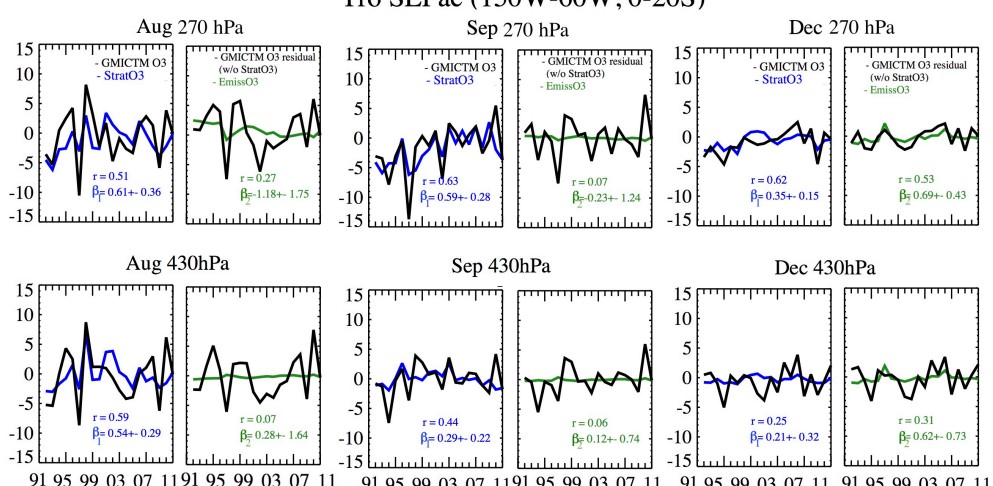

**Figure 16:** The multi-regression results of simulated ozone anomalies over tropical southeastern Pacific region relying on two predictor variables: StratO$_3$ (blue), EmissO$_3$ (green) at 270 hPa and 430 hPa. Three panels show results from August (left), September (middle) and December (right) from 1992 to 2011. Each panel contains two columns. The left column of each panel compares the anomalies of StratO$_3$ (blue) and simulated ozone mixing ratio (black) from the GMI-CTM model at 270 and 430 hPa. The right column compares the simulated O$_3$ residual after removing the regression from StratO$_3$ (black line) and EmissO$_3$ (green line) at these two levels. EmissO$_3$ is calculated from the difference of simulated ozone between the run with yearly-varied emission and the run with constant emission. Unit for y-axis is ppb. The correlation (r), regression coefficient (β) and its 95% confidence level are labeled in each panel.





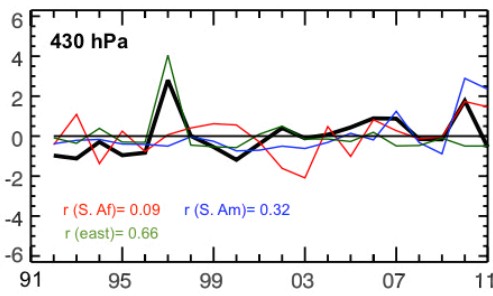


**Figure 17: The standardized anomalies of the tagged CO tracers over tropical southeastern Pacific region from**


**three burning regions, including southern Africa (red), South America (blue) and Eastern region (green) and**


**their comparison with the EmissO3 (black) at 270 and 430 hPa in September from 1992 to 2011.**









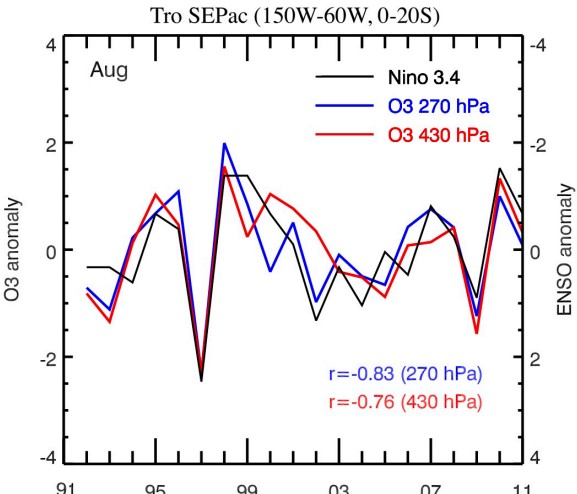


**Figure 18: Comparison of IAV of ozone anomalies over tropical southeastern Pacific region at 270 hPa (blue) and 430 hPa (red) with Niño 3.4 index in August from 1992 to 2011. The 2nd y-axis for the ENSO anomaly is reversed.**



