# Peer review of "Causes of interannual variability over the southern"

_Atmospheric Chemistry and Physics, 2016_

## Referee Comment (RC1) · Anonymous Referee #1 · 14 Nov 2016

The manuscript of Liu et al. discusses the interannual variability of tropospheric ozone over regions where the southern tropospheric ozone maximum is found. This is a well-established feature of tropospheric composition, though such a systematic exploration of its interannual variability in different horizontal and vertical regions, and with a focus on exploring the role of different drivers has not been pursued before. The manuscript is certainly within the scope of ACP, it is generally well written, and the findings will be useful for the understanding of tropospheric ozone variability further. I recommend its publication following some (mostly minor) suggested modifications described below.

GENERAL COMMENT:

If find the second part of the title misleading. The Southern Ocean is mentioned, but this Ocean's northernmost limit is usually taken as 50 or 60S, which is far from where

the focus of this study lies. I suggest modifying possibly to "Causes of interannual variability over the southern hemispheric tropospheric ozone maximum".

SPECIFIC COMMENTS:

Page 2, Line 30: What is special about September, leading to the "even during September" statement. It is not clear at this stage.

Page 2, Line 39: Suggest changing to "especially in the upper troposphere".

Figure 1: Define "upper tropospheric" in the caption.

Page 4, Line 81: Also, Voulgarakis et al. (2011) demonstrated that between transport processes, it is the STE that is the key driver following El Niño events.

It is also worth mentioning somewhere in the introduction that Hess and Mahowald (2009), who prescribed stratospheric ozone, found that IAV of ozone at 500hPa did not show features similar to the Southern Hemisphere ozone maximum described here (see their Fig. 2 & 3), possibly implying the important role of the stratosphere.

Page 5, Line 121: Please change "section" to "Section", as there is only one Section 3.

Page 5, Line 129: Gap after http:// not needed.

Page 5, Line 130: Same amount of levels after re-gridding?

Page 6, Line 136: Please check end of sentence and amend.

Page 6, Lines 142-145: Emissions are important, since their role is investigated, so there needs to be an at least brief mention of what they are here. A quick mention of the reference is not enough. Also: Why was specifically 2000 used for the fixed emissions simulation? Any implications of this selection?

Page 6, Line 148: Mention the global total of lightning emissions again. In fact, this is where the more detailed description of what was used for lightning belongs.

Page 6, Line 151-153: Do they vary with time (e.g. are there any trends in CFCs and

N2O, which would affect ozone)?

Page 6, Line 157: They are both artificial, so please specify that you are referring to e90 (i.e. "The e90 tracer is. . .").

Page 7, Line 168: Why is higher resolution used in this simulation?

Page 9, Line 230: Not clear how the Walker circulation affects the meridional structure of stratospheric ozone contribution, given that the WC occurs in the zonal direction. Maybe the authors mean that the zonal (and not the meridional) variations in the southernmost extent are driven by the WC?

Page 9, Lines 235-237: It is not clear what is suggested here. For ozone in the tropics to be associated with StratO3, I would think that the upper and lower panels of Fig. 2 should have a resemblance in the tropics. That is not something obvious on the figure. Moreover, how can one see an ozone minimum in the three regions mentioned from Figure 2 (upper panel)?

Page 9, Lines 241-242: The Southern Ocean is mentioned, but this Ocean's northernmost limit is usually taken as 50 or 60S, which is far from where the stratospheric influence is found. I suggest changing to "southern Indian and Pacific Oceans".

Page 9, Lines 248-251: Please explain why the southern Pacific was not also selected for study.

Page 10, Lines 254-256: It would have been nice to show a simple map with IAVs. Similar to Fig. 1, but for IAV (e.g. standard deviation divided by the mean). It would give an immediate first view of where the "hot-spots" of variability are, both for certain levels and for UTOC.

Figure 3: Why only from 2005 to 2011 and not for the entire period? Also: The labelling of the x-axis could be made more simple/clear.

Page 10, Lines 258-259: This sentence needs to be moved to the caption, to make

clear what is meant by "anomalies".

Figures 3 & 4: I think "and upper tropospheric ozone column (UTOC, integrated from 500 hPa to the tropopause) anomalies" should be moved earlier in the sentence.

Page 11, Lines 284-285: It would be clearer with IAV maps - as I described above - which areas show larger or smaller IAV.

Page 12, Lines 318-321: Why are the authors mentioning this? Perhaps to suggest that this mechanism is probably responsible for the larger IAV in S. Atl. mentioned earlier, even though IAV in African emissions is small (i.e. there is a remote effect). Please clarify. Also: Perhaps use a clearer term instead of "eastern regions". I believe this is not a standard term. At the very least you can define its borders in this sentence rather than later. Or perhaps use "South and Southeast Asia"? BTW: The later definition on lines 324-325 does not seem to include Australia.

Page 12, Line 340: Where do those percentages of variability "explained" come from?

Page 13, Line 368: "great" -> "greater".

Page 13, Line 369: Paragraph too long. Maybe break it here.

Page 14, Line 391: What does a negative response to ENSO mean here? To the ENSO index?

Page 15, Lines 417-418: From the figure it seems that the "eastern region" is the largest contributor, no?

Page 16, Line 443: "lightning activities" -> "lightning activity".

Page 16, Line 455: "NOX" -> "NOx".

Page 17, Line 475: Somewhat vague statement. Deep convection transports (mixes up) ozone-poor air from near the surface to the UT.

Page 19, Lines 549-550: Suggest rephrasing to "The stratospheric contribution is still

significant at 430 hPa, but drops to less than half of that at 270 hPa".

Page 20, Line 564: Also in Young et al. (2013) (see their Fig. 3).

Page 20, Lines 569-570: Suggest rephrasing to "to the radiative forcing of climate".

REFERENCES:

Hess, P. and Mahowald, N. (2009), Interannual variability in hindcasts of atmospheric chemistry: the role of meteorology, Atmos. Chem. Phys., 9, 5261-5280, doi:10.5194/acp-9-5261-2009.

Voulgarakis, A., Hadjinicolaou, P., and Pyle, J. A. (2011), Increases in global tropospheric ozone following an El Niño event: examining stratospheric ozone variability as a potential driver, Atmos. Sci. Lett., 12, 228–232, doi:10.1002/asl.318.

Young, P. J., Archibald, A. T., Bowman, K. W., Lamarque, J.-F., Naik, V., Stevenson, D. S., Tilmes, S., Voulgarakis, A., Wild, O., Bergmann, D., Cameron-Smith, P., Cionni, I., Collins, W. J., Dalsøren, S. B., Doherty, R. M., Eyring, V., Faluvegi, G., Horowitz, L. W., Josse, B., Lee, Y. H., MacKenzie, I. A., Nagashima, T., Plummer, D. A., Righi, M., Rumbold, S. T., Skeie, R. B., Shindell, D. T., Strode, S. A., Sudo, K., Szopa, S., and Zeng, G. (2013), Pre-industrial to end 21st century projections of tropospheric ozone from the Atmospheric Chemistry and Climate Model Intercomparison Project (ACCMIP), Atmos. Chem. Phys., 13, 2063-2090, doi:10.5194/acp-13-2063-2013.

---

## Referee Comment (RC2) · Anonymous Referee #2 · 12 Dec 2016

Review of Liu et al., Causes of interannual variability of tropospheric ozone over the Southern Ocean

The manuscript by Liu et al. presents an analysis of a series of runs with the Global Modelling Initiative (GMI) CTM driven by MERRA re-analysis to look at the inter-annual variability of ozone in the middle to upper troposphere in regions of the southern hemisphere. To investigate the contribution of stratospheric input on ozone, a diagnostic tracer of stratospheric ozone is included. To estimate the role of inter-annual variability in emissions, the difference between the full simulation and a simulation with constant emissions is used. Multiple linear regression and correlations are used to estimate the contribution of these influences on the year-to-year variability in the model ozone. The study finds a significant contribution of the stratosphere to ozone variability in the upper troposphere, even deep into the tropics, a finding that furthers our evolving understanding of the significant role stratospheric input can have on ozone in the troposphere.

The paper is well written and clearly presents a well thought out analysis. I do not have any significant concerns with the material presented. My one methodological concern is the approach to quantify the contribution of stratospheric ozone (stratO3) and the interannual variability in ozone precursor emissions (emissO3). For example, for the South Atlantic region Figure 6 presents the multiple linear regression (MLR) of stratO3 and emissO3 against the model ozone anomaly. The combination of these two factors can reproduce a high degree of the interannual variability of the model ozone, up to nearly 76% for December at 270 hPa. To separate the contribution of stratO3 and emissO3, the correlation of the stratO3 term from the MLR against the original model ozone timeseries is calculated. Then the contribution of emissO3 is calculated from the correlation of the emissO3 term against the residual that results from removing the stratO3 contribution. During the original MLR analysis the stratO3 and emissO3 terms were simultaneously fitted to the ozone anomaly, but the contribution of stratO3 and emissO3 is calculated by correlation sequentially. The end result is that while the combined stratO3/emissO3 regression explains 76% of the variance for December at 270 hPa (Figure 6), individually stratO3 accounts for 61% and emissO3 accounts for 40% (Figure 7). Given the process of simultaneously fitting the stratO3 and emissO3 terms during the MLR, is not the correct way to calculate their individual contributions to regress these terms individually against the original timeseries? I would argue that if correlation of stratO3 accounts for 61% of the variance, then emissO3 should account for approximately 15% since the combination of the two accounts for 76%. The process seems to work in the extreme where one component explains all of the variance – the south Atlantic at 270 hPa in August, for example – but for cases where both components contribute substantially the approach of regressing the second term against the residual seems to give an inflated estimate. This could be because the process of calculating the residual by removing the contribution from the first term has also removed a large fraction of the variance? And since there is no correct order to which of the two terms is fitted first and which is fitted second, they both should be correlated against

the same (original) timeseries. Following this approach one could argue that emissO3 explains a certain fraction of the residual variance, but one could not directly compare the stratO3 and emissO3 correlations.

The change in methodology argued for above may have some impact on the conclusion of the relative importance of stratO3 and emissO3 for certain regions at certain times but I do not see how it would fundamentally alter the conclusions of the paper.

My other comments are mostly minor and related to specific parts of the paper. They are detailed below.

Lines 103-104. A minor quibble that part of the treatment of lightning NOx is discussed here, where it is stated that the global total is fixed at 5 Tg-N/year, and part is discussed at Lines 146-148. It would help the reader to rework a bit these two parts to combine them in one place.

Lines 103-104. If lightning NOx emissions are held constant, how do you derive the interannual variability in lightning NOx that is used in the correlation shown in Figure 14. It must be the variability over a particular region, but I am not sure I found where that is discussed.

Lines 143-145. I guess it is obvious that the run with constant emissions fixed at the year 2000 levels means that the annual cycle of year 2000 emissions repeats. Sorry for another quibble, but it would help remove any doubt if the wording were more explicit.

Lines 159 - 162. Here the stratO3 tracer is discussed. When it is stated that the stratO3 tracer is 'removed in the troposphere with the same loss frequency...' is that the same loss frequency as Ox and how exactly is Ox defined? Would you know the global average tropospheric O3 lifetime that you would derive from the loss frequency you used for stratO3?

Line 222 . '...represents [the] fraction of tropospheric ozone from [the] stratosphere...'

Line 237. I would suggest removing 'of' from 'Within the Atlantic, despite of the...'

Lines 259 – 261. Here it is mentioned that the interannual variability in the GMI simulation is larger than in the GMAO assimilated ozone for the two tropical regions. Is there any additional information that could be provided as to why this may be the case? Perhaps some comparisons from the Wargan et al. (2015) paper against independent observations or the role of emissions in the assimilation that is mentioned at Lines 277-279? This would seem to be an important component of the comparison if one is to have confidence in the analysis of interannual variability presented later in the paper.

Lines 367 – 369. The statement on the relative contribution of emissions to ozone variability at 270 and 430 hPa will probably need to be revisited if the method of attribution is revised as argued for above.

Lines 454-456. On Figure 12, it would be interesting to see the same fit of ozone with lightning at 430 hPa as is shown for 267 hPa.

Line 484-485. 'Figure 14 compares the model residual after removing the contributions from StratO3 and EmissO3...' and I would raise the same concern that the analysis is overestimating the contribution of lightning to explaining the variance in ozone.

Lines 552 – 556. Because the correlation of lightning with ozone variability is negative, the authors suggest deep convection is having a negative effect on ozone in the upper troposphere by lofting clean surface air. I agree that could definitely be a possibility, but can you rule out that the correlation is signalling some other effect? Perhaps circulation changes that are associated with the interannual variability in deep convection?

Lines 825-829. The colour scale on Figure 1 indicates it is ppb and it should be DU as I understand it.

---

## Author Comment (AC1) · 2 Feb 2017

Final author comments on "Causes of interannual variability of tropospheric ozone over the Southern Ocean" by Junhua Liu et al.

We thank the two reviewers for their comments. Both of them recommend publication with minor revisions. We have addressed all comments in detail below and have clarified the text in the relevant sections.

In the following, we address the concerns raised by both reviewers. Reviewers' comments are italicized.

*Anonymous Referee #1*

*The manuscript of Liu et al. discusses the interannual variability of tropospheric ozone over regions where the southern tropospheric ozone maximum is found. This is a well-established feature of tropospheric composition, though such a systematic exploration of its interannual variability in different horizontal and vertical regions, and with a focus on exploring the role of different drivers has not been pursued before. The manuscript is certainly within the scope of ACP, it is generally well written, and the findings will be useful for the understanding of tropospheric ozone variability further. I recommend its publication following some (mostly minor) suggested modifications described below.*

*GENERAL COMMENT:*
*1: I find the second part of the title misleading. The Southern Ocean is mentioned, but this Ocean's northernmost limit is usually taken as 50 or 60S, which is far from where the focus of this study lies. I suggest modifying possibly to "Causes of interannual variability over the southern hemispheric tropospheric ozone maximum".*

The title has been modified as suggested in the revised manuscript.

*SPECIFIC COMMENTS:*
*2: Page 2, Line 30: What is special about September, leading to the "even during September" statement. It is not clear at this stage.*

September is the month that CO has the largest contribution from southern hemispheric biomass burning. We deleted 'even during September' to make the context clear.

*3: Page 2, Line 39: Suggest changing to "especially in the upper troposphere".*
The text has been modified as suggested

*4: Figure 1: Define "upper tropospheric" in the caption.*
The definition of "upper tropospheric" has been added in the caption.

*5: Page 4, Line 81: Also, Voulgarakis et al. (2011) demonstrated that between transport processes, it is the STE that is the key driver following El Niño events. It is also worth mentioning somewhere in the introduction that Hess and Mahowald (2009), who prescribed stratospheric ozone, found that IAV of ozone at 500hPa did not show features*

*similar to the Southern Hemisphere ozone maximum described here (see their Fig. 2 & 3), possibly implying the important role of the stratosphere.*

These two references have been added in the text. Please see below:

Voulgarakis et al. (2011) demonstrated that increases in the amounts of stratospheric ozone entering the troposphere following El Niño events are mainly driven by changes in the STE.

Hess and Mahowald (2009) used a CTM to quantify relative interannual variability in global model ozone in hindcast simulations with constant emissions and prescribed stratospheric ozone. The CTM was driven by two sets of meteorological fields: a) the National Center for Environmental Prediction/National Center for Atmospheric Research reanalysis; b) from a simulation using the Community Atmosphere Model (CAM-3) forced with observed sea surface temperatures. Their study found that relative IAV of ozone at 500 hPa shows the maximum between the Equator and 30S in JJA and DJF.

*6: Page 5, Line 121: Please change "section" to "Section", as there is only one Section 3.*
The text has been modified as suggested.

*7: Page 5, Line 129: Gap after http:// not needed.*
The text has been modified as suggested.

*8: Page 5, Line 130: Same amount of levels after re-gridding?*
Yes, the vertical levels remain unchanged. The text has been modified as: we regrid it to 2°x2.5° horizontal grid for input to the GMI-CTM simulations in this study.

*9: Page 6, Line 136: Please check end of sentence and amend.*
The text has been modified.

*10: Page 6, Lines 142-145: Emissions are important, since their role is investigated, so there needs to be an at least brief mention of what they are here. A quick mention of the reference is not enough.*

The text has been modified as below:

The GMI-CTM standard simulation (labeled as Hindcast-VE) used in this study for 1992-2011 includes monthly and inter-annually varying emissions with anthropogenic, biomass burning, and biogenic sources. Anthropogenic emissions are based on the EDGAR 3.2 Inventory (Olivier et al., 2005), overwritten with available regional inventories for North America, Europe, Asia and Mexico. More details are given in Strode et al. (2015). Biomass burning emissions are from the Global Fire Emission Database, GFED3 (van der Werf et al., 2010). Emission before 1997 are obtained from GFED3 emission climatology averaged for 2001 to 2009 applied with regional-scale IAV, which was derived from satellite information on fire activity (ATSR) and/or aerosol optical depths

from the Total Ozone Mapping Spectrometer (TOMS) by Duncan et al. (2003). Biogenic emissions of isoprene and monoterpenes follow the latest version of the MEGAN algorithm (Guenther et al., 2006). Besides the standard simulation, we carry out a control run with anthropogenic and biomass emissions fixed at year 2000 levels. The comparison between the control and standard simulation allows us to quantify effects of emission IAV on ozone IAV.

*Also: Why was specifically 2000 used for the fixed emissions simulation? Any implications of this selection?*

Year 2000 is about the middle point of examined period (1991-2011). However, the selection of year with constant emission does not affect the conclusion of how emission IAV affects the ozone IAV.

*11: Page 6, Line 148: Mention the global total of lightning emissions again. In fact, this is where the more detailed description of what was used for lightning belongs.*

We moved the lightning description in the Introduction Section to here as below:
In our GMI-CTM, the lightning parameterization follows the scheme described by Allen et al (2010). The regional lightning $NO_X$ emission, calculated online by coupling to the deep convective transport in the model, varies from year to year. The global total of $NO_X$ from lightning is fixed at 5.0 TgN/yr.

*12: Page 6, Line 151-153: Do they vary with time (e.g. are there any trends in CFCs and N2O, which would affect ozone)?*

Yes, both CFCs and $N_2O$ have trends. CFCs increase before 1999 then decrease after 1999. $N_2O$ shows an increasing trend through the study period. The trend in stratospheric ozone due to CFCs and $N_2O$ during the period of interest is small compared with IAV in stratospheric ozone input to the troposphere. Meanwhile, in this study, we examined the effect of IAV of ozone input from stratosphere on the IAV of tropospheric ozone using the $StratO_3$ tracer. The variations of the contribution from stratospheric ozone could result from the variation in stratospheric ozone (which is relate to variations in CFCs and $N_2O$) or the changes in STE or both. We did not separate these effects in our study. Further examination of their separate effects is beyond the scope of this paper.

*13: Page 6, Line 157: They are both artificial, so please specify that you are referring to e90 (i.e. "The e90 tracer is...").*
The text has been modified as suggested.

*14: Page 7, Line 168: Why is higher resolution used in this simulation?*
We did our analysis using the highest resolution that available in our simulations.

*15: Page 9, Line 230: Not clear how the Walker circulation affects the meridional structure of stratospheric ozone contribution, given that the WC occurs in the zonal*

*direction. Maybe the authors mean that the zonal (and not the meridional) variations in the southernmost extent are driven by the WC?*
Two places in this paragraph have been changed into 'zonal circulation'.

*16: Page 9, Lines 235-237: It is not clear what is suggested here. For ozone in the tropics to be associated with StratO3, I would think that the upper and lower panels of Fig. 2 should have a resemblance in the tropics. That is not something obvious on the figure. Moreover, how can one see an ozone minimum in the three regions mentioned from Figure 2 (upper panel)?*

We modified the text as suggested in places where clarification was needed. Please see line 259-272 in the modified manuscript.

The bottom panel of Figure 2 suggests the regions with minimum stratospheric ozone contribution in the tropics reaches further south over Indian Ocean than the tropical eastern Pacific and Atlantic. The zonal variation of stratospheric contribution in the tropics is in agreement with that of ozone as shown in the upper panel of Figure 2, showing elevated ozone over the tropical eastern Pacific and Atlantic and the minimum over Indian Ocean.

*17: Page 9, Lines 241-242: The Southern Ocean is mentioned, but this Ocean's northernmost limit is usually taken as 50 or 60S, which is far from where the stratospheric influence is found. I suggest changing to "southern Indian and Pacific Oceans".*
The text has been modified as suggested.

*18: Page 9, Lines 248-251: Please explain why the southern Pacific was not also selected for study.*

We did not select southern Pacific is because: although tropospheric ozone is elevated over this region, it does not reach the regional maximum. The ozone concentration in this region is lower than that in southern Atlantic and southern Indian Ocean. We did a similar analysis on controlling factors of tropospheric ozone over the southern Pacific. Our results suggest that stratospheric ozone input playing a dominant role on the IAV of tropospheric ozone over southern Pacific.

*19: Page 10, Lines 254-256: It would have been nice to show a simple map with IAVs. Similar to Fig. 1, but for IAV (e.g. standard deviation divided by the mean). It would give an immediate first view of where the "hot-spots" of variability are, both for certain levels and for UTOC.*
We added Figure R1 into modified manuscript as Figure 3.

[Figure]

Figure R1: The IAV of simulated ozone at 270 hPa (top) and 430 hPa (bottom). The IAV is represented by the standard deviation of ozone anomalies (removing the monthly mean) over 1991-2011. Stronger ozone IAV happens over subtropical south Atlantic and subtropical south Indian Ocean at 270 hPa. At 430 hPa, Tropical southeastern Pacific and tropical South Atlantic has slightly larger IAV.

*20: Figure 3: Why only from 2005 to 2011 and not for the entire period? Also: The labeling of the x-axis could be made more simple/clear.*
We modified the x-label for Figure 4 and 5 in the revised manuscript. Aura data are only available since late 2004.

*21: Page 10, Lines 258-259: This sentence needs to be moved to the caption, to make clear what is meant by "anomalies".*
This sentence has been added to the caption.

*22: Figures 3 & 4: I think "and upper tropospheric ozone column (UTOC, integrated from 500 hPa to the tropopause) anomalies" should be moved earlier in the sentence. Page 11, Lines 284-285: It would be clearer with IAV maps - as I described above - which areas show larger or smaller IAV.*
We modified the figures and captions to show tropospheric column situation first.
For Line 284-285, we modified the text to be precise.

*23: Page 12, Lines 318-321: Why are the authors mentioning this? Perhaps to suggest that this mechanism is probably responsible for the larger IAV in S. Atl. mentioned earlier, even though IAV in African emissions is small (i.e. there is a remote effect). Please clarify. Also: Perhaps use a clearer term instead of "eastern regions". I believe this is not a standard term. At the very least you can define its borders in this sentence*

*rather than later. Or perhaps use "South and Southeast Asia"? BTW: The later definition on lines 324-325 does not seem to include Australia.*

Emissions from South and Southeast Asia affect the southern hemisphere along with emissions from Africa and South America. We therefore include this region in our discussion.

The larger IAV in S. Atlantic results from the larger IAV from South America biomass burning.

We clarified the definition of eastern region in the text and replaced the eastern region with 'South and Southeast Asia" both in text and figure.

*24: Page 12, Line 340: Where do those percentages of variability "explained" come from?*
These are calculated from the correlations shown in the figure 7 in the revised manuscript.

*25: Page 13, Line 368: "great" -> "greater".*

The text has been modified as suggested.

*26: Page 13, Line 369: Paragraph too long. Maybe break it here.*
The paragraph has been modified as suggested.

*27: Page 14, Line 391: What does a negative response to ENSO mean here? To the ENSO index?*

We replaced the "ENSO" with "the Niño 3.4 index".

*28: Page 15, Lines 417-418: From the figure it seems that the "eastern region" is the largest contributor, no?*

The discussion here is for the situation at 430 hPa in September, which is the left bottom panel of Figure 12 in the revised manuscript. The emissions from S. America and southern Africa are the larger emission contributors at 430 hPa in September. In the next few lines, we mentioned that emission from South and Southeast Asia is the largest contributor in December at both levels.

*29: Page 16, Line 443: "lightning activities" -> "lightning activity".*
The text has been modified as suggested.

*30: Page 16, Line 455: "NOX" -> "NOx".*
The text has been modified as suggested.

*31: Page 17, Line 475: Somewhat vague statement. Deep convection transports (mixes up) ozone-poor air from near the surface to the UT.*

The text has been modified. Please see below.
Deep convection over a clean region reduces upper tropospheric ozone by mixing up ozone-poor air from near the surface. This effect could be opposite if deep convection happens over a polluted region with relatively high ozone and its precursors (Lawrence et al., 2003; Ziemke, et al., 2015).

*32: Page 19, Lines 549-550: Suggest rephrasing to "The stratospheric contribution is still significant at 430 hPa, but drops to less than half of that at 270 hPa".*
The text has been modified as suggested.

*33: Page 20, Line 564: Also in Young et al. (2013) (see their Fig. 3).*
The reference has been added as suggested.

*34: Page 20, Lines 569-570: Suggest rephrasing to "to the radiative forcing of climate".*
The text has been modified as suggested.
*Review of Liu et al., Causes of interannual variability of tropospheric ozone over the Southern Ocean*

*The manuscript by Liu et al. presents an analysis of a series of runs with the Global Modelling Initiative (GMI) CTM driven by MERRA re-analysis to look at the inter-annual variability of ozone in the middle to upper troposphere in regions of the southern hemisphere. To investigate the contribution of stratospheric input on ozone, a diagnostic tracer of stratospheric ozone is included. To estimate the role of inter-annual variability in emissions, the difference between the full simulation and a simulation with constant emissions is used. Multiple linear regression and correlations are used to estimate the contribution of these influences on the year-to-year variability in the model ozone. The study finds a significant contribution of the stratosphere to ozone variability in the upper troposphere, even deep into the tropics, a finding that furthers our evolving understanding of the significant role stratospheric input can have on ozone in the troposphere.*

*The paper is well written and clearly presents a well thought out analysis. I do not have any significant concerns with the material presented.*
*1: My one methodological concern is the approach to quantify the contribution of stratospheric ozone (stratO3) and the interannual variability in ozone precursor emissions (emissO3). For example, for the South Atlantic region Figure 6 presents the multiple linear regression (MLR) of stratO3 and emissO3 against the model ozone anomaly. The combination of these two factors can reproduce a high degree of the interannual variability of the model ozone, up to nearly 76% for December at 270 hPa. To separate the contribution of stratO3 and emissO3, the correlation of the stratO3 term from the MLR against the original model ozone timeseries is calculated. Then the contribution of emissO3 is calculated from the correlation of the emissO3 term against the residual that results from removing the stratO3 contribution. During the original MLR analysis the stratO3 and emissO3 terms were simultaneously fitted to the ozone anomaly, but the contribution of stratO3 and emissO3 is calculated by correlation sequentially. The end result is that while the combined stratO3/emissO3 regression explains 76% of the variance for December at 270 hPa (Figure 6), individually stratO3 accounts for 61% and emissO3 accounts for 40% (Figure 7).*
*Given the process of simultaneously fitting the stratO3 and emissO3 terms during the MLR, is not the correct way to calculate their individual contributions to regress these terms individually against the original timeseries? I would argue that if correlation of stratO3 accounts for 61% of the variance, then emissO3 should account for approximately 15% since the combination of the two accounts for 76%. The process seems to work in the extreme where one component explains all of the variance – the south Atlantic at 270 hPa in August, for example – but for cases where both components contribute substantially the approach of regressing the second term against the residual seems to give an inflated estimate. This could be because the process of calculating the*

*residual by removing the contribution from the first term has also removed a large fraction of the variance? And since there is no correct order to which of the two terms is fitted first and which is fitted second, they both should be correlated against the same (original) timeseries. Following this approach one could argue that emissO3 explains a certain fraction of the residual variance, but one could not directly compare the stratO3 and emissO3 correlations.*
*The change in methodology argued for above may have some impact on the conclusion of the relative importance of stratO3 and emissO3 for certain regions at certain times but I do not see how it would fundamentally alter the conclusions of the paper.*

Thanks a lot for the reviewer's comments on this issue. We agree with the comments and modified the calculated as suggested.

Considering that the regressors ($StratO_3$, $EmisO_3$, lightningNOx) might be correlated and not orthogonal with each other, we estimate the amount of variance explained by each regressor following the method described in ( Kruskal, 1987;Chevan and Sutherland, 1991; Groemping, 2007). In this method, regressor is added to the model one by one and the corresponding sequential sum of squares for each regressor is calculated. The sequential sum of squares depends on the regressors already in the model; we therefore do the calculation for every possible order in which regressors can enter the model, and then average over orders. Below are two examples of variance table.
1) The first one is for the multi-regression with two regressors

Table 1: Analysis of variance for regression with $StratO_3$ and $EmisO_3$ over South Atlantic in December at 270 hPa.

| Source | SS |
|---|---|
| Regression | 135.94 |
| Error | 45.47 |
| Total | 181.41 |
| Variance by regression | 0.75 |

Sequential sum of square

| Source | Seq SS (StraO3) | Seq SS (EmisO3) |
|---|---|---|
| StratO3 + EmisO3 | 110.81 | 25.13 |
| EmisO3 + StratO3 | 59.29 | 76.65 |
| | | |
| Source | StratO3 | EmisO3 |
| Mean SS | 85.05 | 50.89 |
| Variance explained | 0.47 | 0.28 |

2) The second one is for the multi-regression with three regressors ($StratO_3$, $EmisO_3$, and lightning NOx) over tropical Atlantic in September at 270 hPa

Table 2: Analysis of variance table for regression with $StratO_3$, $EmisO_3$ and lightning NOx over Tropical Atlantic in September at 270 hPa.

| Source | SS |
|---|---|
| Regression | 210.05 |
| Error | 108.98 |
| Total | 319.03 |
| Variance by regression | 0.66 |

Sequential sum of square

| Source | Seq SS (StraO$_3$) | Seq SS (EmisO$_3$) | Seq SS (lightningNOx) |
|---|---|---|---|
| StratO$_3$ + EmisO$_3$ + LightningNOx | 167.25 | 7.02 | 35.78 |
| StratO$_3$ + LightningNOx + EmisO$_3$ | 167.25 | 0.88 | 41.92 |
| EmisO$_3$ + StratO$_3$ + LightningNOx | 156.08 | 18.19 | 35.78 |
| EmisO$_3$ + LightningNOx + StratO$_3$ | 112.90 | 18.19 | 78.96 |
| LightningNOx + StratO$_3$ + EmisO$_3$ | 114.24 | 0.88 | 94.94 |
| LightningNOx + EmisO$_3$ + StratO$_3$ | 112.90 | 2.21 | 94.94 |

| Source | StratO$_3$ | EmisO$_3$ | Lightning NOx |
|---|---|---|---|
| Mean SS | 138.44 | 7.90 | 63.72 |
| Variance explained | 0.43 | 0.03 | 0.20 |

We modified the discussion in the text. The changes in methodology discussed above have impact on the value of the relative contributions of stratO$_3$ and emissO$_3$ for certain regions at certain times, but the conclusion of relative importance does not change.

*My other comments are mostly minor and related to specific parts of the paper. They are detailed below.*

*2: Lines 103-104. A minor quibble that part of the treatment of lightning NOx is discussed here, where it is stated that the global total is fixed at 5 Tg-N/year, and part is discussed at Lines 146-148. It would help the reader to rework a bit these two parts to combine them in one place.*

The text has been modified. Please see our response to question 11 of reviewer 1

*3: Lines 103-104. If lightning NOx emissions are held constant, how do you derive the interannual variability in lightning NOx that is used in the correlation shown in Figure 14. It must be the variability over a particular region, but I am not sure I found where that is discussed.*

In the modified manuscript, we mentioned how regional lightning NOx is calculated online. Please see below: The regional NO$_X$ emission from lightning is calculated online by coupling to the deep convective transport in the model and varies from year to year.

*4: Lines 143-145. I guess it is obvious that the run with constant emissions fixed at the year 2000 levels means that the annual cycle of year 2000 emissions repeats. Sorry for another quibble, but it would help remove any doubt if the wording were more explicit.*
Yes. The text has been modified for clarification.

Besides the standard simulation, we carry out a control run with anthropogenic and biomass emissions hold at year 2000 level with seasonality.

*5: Lines 159 - 162. Here the stratO3 tracer is discussed. When it is stated that the stratO3 tracer is 'removed in the troposphere with the same loss frequency...' is that the same loss frequency as Ox and how exactly is Ox defined? Would you know the global average tropospheric O3 lifetime that you would derive from the loss frequency you used for stratO3?*

The stratosphere $O_3$ is the same with daily output of the respective full chemistry run. The tropopause is defined as e90 tracer to be 75 ppb. The three chemical loss rates in the troposphere are archived from monthly full chemistry run.
$O_1D + H_2O = 2\ OH$
$HO_2 + O_3 = 2\ O_2 + OH$
$OH + O_3 = HO_2 + O_2$
The $StratO_3$ was removed at the surface level, which is equal to the dry deposition process. There is no chemical production of $StratO_3$ in the troposphere.

*6: Line 222 . '...represents [the] fraction of tropospheric ozone from [the] stratosphere...'*

The text has been modified as suggested.

*7: Line 237. I would suggest removing 'of' from 'Within the Atlantic, despite of the...'*
The text has been modified as suggested.

*8: Lines 259 – 261. Here it is mentioned that the interannual variability in the GMI simulation is larger than in the GMAO assimilated ozone for the two tropical regions. Is there any additional information that could be provided as to why this may be the case? Perhaps some comparisons from the Wargan et al. (2015) paper against independent observations or the role of emissions in the assimilation that is mentioned at Lines 277-279? This would seem to be an important component of the comparison if one is to have confidence in the analysis of interannual variability presented later in the paper.*

There are limitations in the assimilation data including 1) No chemistry and lack of emissions in the troposphere in the assimilation, 2) no direct observational constraint in the troposphere. Both could contribute to the less IAV in the GMAO assimilated data at one pressure level in the troposphere, especially in the middle and lower troposphere. For the upper tropospheric column comparison, the agreement in the magnitude of IAV between GMI simulation and GMAO assimilated data improves.

*9: Lines 367 – 369. The statement on the relative contribution of emissions to ozone variability at 270 and 430 hPa will probably need to be revisited if the method of attribution is revised as argued for above.*

Please see our response to question 1 above.

*10: Lines 454-456. On Figure 12, it would be interesting to see the same fit of ozone with lightning at 430 hPa as is shown for 267 hPa.*

With the source originated from the upper troposphere, the lightning NOx has the largest effect in the upper troposphere and the effects are insignificant at 430 hPa. We therefore did not show the comparison at 430 hPa.

*11: Line 484-485. 'Figure 14 compares the model residual after removing the contributions from StratO3 and EmissO3...' and I would raise the same concern that the analysis is overestimating the contribution of lightning to explaining the variance in ozone.*

Please see our response to question 1 above.

*Lines 552 – 556. Because the correlation of lightning with ozone variability is negative, the authors suggest deep convection is having a negative effect on ozone in the upper troposphere by lofting clean surface air. I agree that could definitely be a possibility, but can you rule out that the correlation is signalling some other effect? Perhaps circulation changes that are associated with the interannual variability in deep convection?*

We cannot rule out other possibilities. The reason we focus on convection effects is that in the model, the lightning parameterization is coupling to the deep convective transport. Increase in deep convection produces more upper tropospheric NOx from lightning, which results more ozone production. On the other hand, deep convection could decrease the upper tropospheric ozone by mixing up ozone-poor air from surface. Therefore, the convection has two opposite but quite important and direct effects on upper tropospheric ozone.

*Lines 825-829. The colour scale on Figure 1 indicates it is ppb and it should be DU as I understand it.*

We modified the unit for color bar.

Reference:

Allen, D., Pickering, K., Duncan, B., and Damon, M.: Impact of lightning NO emissions on North American photochemistry as determined using the Global Modeling Initiative (GMI) model, Journal of Geophysical Research-Atmospheres, 115, 10.1029/2010jd014062, 2010.

Chevan, A., and Sutherland, M.: HIERARCHICAL PARTITIONING, American Statistician, 45, 90-96, 10.2307/2684366, 1991.

Duncan, B. N., Martin, R. V., Staudt, A. C., Yevich, R., and Logan, J. A.: Interannual and seasonal variability of biomass burning emissions constrained by satellite

observations, Journal of Geophysical Research-Atmospheres, 108, 10.1029/2002jd002378, 2003.

Groemping, U.: Two simple estimators of relative importance in linear regression based on variance decomposition - Response, American Statistician, 61, 282-283, 2007.

Guenther, A., Karl, T., Harley, P., Wiedinmyer, C., Palmer, P. I., and Geron, C.: Estimates of global terrestrial isoprene emissions using MEGAN (Model of Emissions of Gases and Aerosols from Nature), ATMOS CHEM PHYS, 6, 3181-3210, 2006.

Hess, P., and Mahowald, N.: Interannual variability in hindcasts of atmospheric chemistry: the role of meteorology, ATMOS CHEM PHYS, 9, 5261-5280, 10.5194/acp-9-5261-2009, 2009.

Kruskal, W.: RELATIVE IMPORTANCE BY AVERAGING OVER ORDERINGS, American Statistician, 41, 6-10, 10.2307/2684310, 1987.

Lawrence, M. G., von Kuhlmann, R., Salzmann, M., and Rasch, P. J.: The balance of effects of deep convective mixing on tropospheric ozone, Geophysical Research Letters, 30, 10.1029/2003gl017644, 2003.

Olivier, J. G. J., Van Aardenne, J. A., Dentener, F. J., Pagliari, V., Ganzeveld, L. N., and Peters, J. A. H. W.: Recent trends in global greenhouse gas emissions: regional trends and spatial distribution of key sources, in: Non-CO2 Greenhouse Gases (NCGG-4), coordinator: van Amstel, A., Millpress, Rotterdam, ISBN 905966 043 9, 325–330,, 2005.

Strode, S. A., Rodriguez, J. M., Logan, J. A., Cooper, O. R., Witte, J. C., Lamsal, L. N., Damon, M., Van Aartsen, B., Steenrod, S. D., and Strahan, S. E.: Trends and variability in surface ozone over the United States, Journal of Geophysical Research-Atmospheres, 120, 9020-9042, 10.1002/2014jd022784, 2015.

van der Werf, G. R., Randerson, J. T., Giglio, L., Collatz, G. L., Mu, M., Kasibhatla, P. S., Morton, D. C., DeFries, R. S., Jin, Y., and T.T., v. L.: Global fire emissions and the contribution of deforestation, savanna, forest, agricultural, and peat fires (1997–2009), ATMOS CHEM PHYS, 10, 16153-16230, 2010.

Voulgarakis, A., Hadjinicolaou, P., and Pyle, J. A.: Increases in global tropospheric ozone following an El Nino event: examining stratospheric ozone variability as a potential driver, Atmospheric Science Letters, 12, 228-232, 10.1002/asl.318, 2011.

Young, P. J., Archibald, A. T., Bowman, K. W., Lamarque, J.-F., Naik, V., Stevenson, D. S., Tilmes, S., Voulgarakis, A., Wild, O., Bergmann, D., Cameron-Smith, P., Cionni, I., Collins, W. J., Dalsøren, S. B., Doherty, R. M., Eyring, V., Faluvegi, G., Horowitz, L. W., Josse, B., Lee, Y. H., MacKenzie, I. A., Nagashima, T., Plummer, D. A., Righi, M., Rumbold, S. T., Skeie, R. B., Shindell, D. T., Strode, S. A., Sudo, K., Szopa, S., and Zeng, G. (2013), Pre-industrial to end 21st century projections of tropospheric ozone from the Atmospheric Chemistry and Climate Model Intercomparison Project (ACCMIP), Atmos. Chem. Phys., 13, 2063-2090, doi:10.5194/acp-13-2063-2013.

Ziemke, J.R., A.R. Douglass, L.D. Oman, S.E. Strahan, and B.N. Duncan, "Tropospheric ozone variability in the tropics from ENSO to MJO and shorter timescales", Atmos. Chem. Phys. Discuss., 15, 6373-6401, doi:10.5194/acpd-15-6373-2015, 2015.